# Recent Advanced Supercapacitor: A Review of Storage Mechanisms, Electrode Materials, Modification, and Perspectives

**DOI:** 10.3390/nano12203708

**Published:** 2022-10-21

**Authors:** Niraj Kumar, Su-Bin Kim, Seul-Yi Lee, Soo-Jin Park

**Affiliations:** Department of Chemistry, Inha University, Incheon 22212, Korea

**Keywords:** carbon-based materials, metal oxides, conductive polymers, hybrid capacitors, supercapacitors

## Abstract

In recent years, the development of energy storage devices has received much attention due to the increasing demand for renewable energy. Supercapacitors (SCs) have attracted considerable attention among various energy storage devices due to their high specific capacity, high power density, long cycle life, economic efficiency, environmental friendliness, high safety, and fast charge/discharge rates. SCs are devices that can store large amounts of electrical energy and release it quickly, making them ideal for use in a wide range of applications. They are often used in conjunction with batteries to provide a power boost when needed and can also be used as a standalone power source. They can be used in various potential applications, such as portable equipment, smart electronic systems, electric vehicles, and grid energy storage systems. There are a variety of materials that have been studied for use as SC electrodes, each with its advantages and limitations. The electrode material must have a high surface area to volume ratio to enable high energy storage densities. Additionally, the electrode material must be highly conductive to enable efficient charge transfer. Over the past several years, several novel materials have been developed which can be used to improve the capacitance of the SCs. This article reviews three types of SCs: electrochemical double-layer capacitors (EDLCs), pseudocapacitors, and hybrid supercapacitors, their respective development, energy storage mechanisms, and the latest research progress in material preparation and modification. In addition, it proposes potentially feasible solutions to the problems encountered during the development of supercapacitors and looks forward to the future development direction of SCs.

## 1. Introduction

In recent years, the world has experienced an increase in development, leading to energy shortages and global warming. These problems have underscored the need for supercapacitors as green energy storage devices. Supercapacitors can store large amounts of energy and deliver excellent power, making them ideal for various applications. Supercapacitors are an increasingly attractive option in the race to develop new and improved energy storage technologies due to their high-power density and long cycle life. As the supercapacitor market grows, so does the need for improved fabrication processes and electrode materials. Supercapacitors have several advantages over other energy storage devices. They can charge and discharge quickly, making them well-suited for various applications. In addition, supercapacitors are environmentally friendly and have a long lifetime. Supercapacitors are expected to grow in the coming years as the world looks for ways to address energy shortages and global warming.

Identifying clean and renewable new energy sources and developing efficient energy storage technologies and devices for low-carbon and sustainable economic development have become important [1,2,3,4]. Common electrochemical energy storage and conversion systems include batteries, capacitors, and supercapacitors [5]. The three energy storage systems complement each other in practical applications and meet different needs in different situations. Although the three systems have different energy storage and conversion mechanisms, they are all based on similar electrochemical thermodynamics and kinetics, i.e., the process of supplying energy occurs at the phase boundary of the electrode/electrolyte interface with independent electron and ion transport [6]. Recent advances in smart electronic devices have spurred a corresponding increase in the use of supercapacitors.

A supercapacitor is a promising energy storage device between a traditional physical capacitor and a battery. Based on the differences in energy storage models and structures, supercapacitors are generally divided into three categories: electrochemical double-layer capacitors (EDLCs), redox electrochemical capacitors (pseudocapacitors), and hybrid capacitors (Figure 1) [7]. Figure 1 summarizes the basic energy storage principles of supercapacitors with the classification as the basic framework and examines the research progress of electrode materials commonly used in recent years. 

Supercapacitors are being researched extensively in smart electronics applications such as flexible, biodegradable, transparent, wearable, flexible, on-chip, and portable energy storage. In comparison with conventional capacitors, supercapacitors use materials with a high specific surface area as electrodes [8,9]. A higher specific surface area and thinner dielectrics result in greater specific capacitance and energy density. In comparison with the rated capacitance of traditional capacitors in the range between micro and milli- Farads, the capacitance of a supercapacitor unit can reach thousands of Farads. In contrast with batteries, the charge storage mechanism of supercapacitors is based on the surface reaction of the electrode material, and there is no diffusion of ions inside the material. Therefore, supercapacitors have a better power density under the same volume. Another electrochemical characteristic that is different between supercapacitors and batteries is that the charge on the electrodes of a typical supercapacitor always increases (or decreases) linearly, resulting in voltage rise (or fall) during the charge and discharge process.

As shown in Figure 2, during the charge and discharge process, the cyclic voltammetry (CV) curve of the supercapacitor (Figure 2a) remains rectangular, whereas the current is almost constant. In addition, its galvanostatic charge-discharge (GCD) curve (Figure 2c) is usually inclined with a constant slope. A battery exhibits Faradaic reactions during the charge and discharge process, and its CV curve shows a clear redox peak; it maintains a constant voltage except when it is near 100% charged/discharged (TOC/EOD) (the GCD curve shows a relatively flat charge-discharge platform).

Supercapacitors have many other advantages, such as being environmentally friendly, having a long service life, being able to operate in wide temperature ranges, and being good at retaining charge even when large currents are applied; they are widely used in consumer electronics, smart meters, and transport [7,8,9,10]. Supercapacitors have shown that they can perform very well in some applications, but there are still several shortcomings that are relevant in some applications. These include high energy density requirements and very long working times. Notably, the areas in which supercapacitors can store charge are limited to the surfaces (or near the active surface area) of the electrodes; they have lower energy density than batteries. 

According to the energy density formula $E=\frac{1}{2}CV^{2}$ (E is the energy density, C is the specific capacitance, and V is the voltage window), the energy density of a capacitor depends on the specific capacitance of the electrode material and the potential difference between the positive and negative electrodes. One of the most effective ways to increase the energy density of supercapacitors is to develop porous nanoelectrode materials. By increasing the specific surface area, porous nanoelectrode materials can increase the specific capacitance and, thus, the energy density. This is a highly effective way to improve the performance of supercapacitors and has the potential to revolutionize the way they are used in a variety of applications. Another method is building hybrid/asymmetric supercapacitors, which can increase the overall device performance.

In this review, we review the recent progress in electrode materials for supercapacitors. Supercapacitors are becoming indispensable in modern electronics with several promising advantages. Therefore, it is necessary to deeply understand and thoroughly summarize the recent progress and development in electrode materials for supercapacitors. This review describes the recent advances and development in electrode materials for supercapacitor devices. Moreover, it briefly explains the mechanisms of supercapacitor storage, the potential applications of this technology, and the challenges associated with its development. This review has a comprehensive overview of the characteristics of types of supercapacitors. It briefly explains the latest research with potential research gaps and the types of electrode materials for supercapacitor electrodes with recent trends and comparative logical perspectives. It also gives some of the research work that has been performed in the past on supercapacitor electrodes and devices. Then, it discusses the future challenges posed by developing electrode materials for supercapacitor applications.

### 1.1. Progress and Classification

SCs are similar in their structure to conventional dielectric capacitors, except that instead of metal, porous electrodes coated with electrolytes to make them more effective, a suitable electrolyte material is used instead of a suitable dielectric to make them suitable for ion channeling. According to different energy storage mechanisms, supercapacitors can generally be divided into EDLCs and pseudocapacitors (Figure 3) [11]. 

#### 1.1.1. EDLC (Electrochemical Double-Layer Capacitors)

EDLCs are currently the most established energy storage device widely used in commercial applications. H. I. Becker (General Electric Company) first demonstrated double-layer capacitance in 1957 and patented this. Later, the first commercial supercapacitor was manufactured in 1971 by Standard Oil Company of Ohio and used in memory applications by a Japanese corporation called NEC. In 1971, it was reported that a new type of capacitor called a pseudocapacitor that used processes of chemical reaction known as Faradaic reactions was developed based on RuO_2_. The finding of pseudocapacitance is exciting as it opens up a new way to improve the charge-storage capabilities of electrochemistry capacitors. The next year, Pinnacle Research Institute (1982) utilized ruthenium oxide as the electrode material to develop its EC devices. They named the device a “pseudo-capacitor” to emphasize the very high performance of the EC devices. The timeline for ECs development is illustrated in Figure 1. EDLCs can store charges through static electricity or non-Faraday processes, which do not involve charge transfer between the electrodes and electrolytes [12]. Therefore, the EDLC storage mechanism allows for rapid energy absorption and transmission and improves power performance.

Due to the absence of Faraday processes, the swelling of the active material during the charge and discharge process of the battery is eliminated, contributing to the excellent cyclic stability of EDLCs. Furthermore, the performance of EDLCs can be modulated according to the type of electrolyte used. However, due to the electrostatic surface charging mechanism, the energy density of EDLC equipment is limited, which greatly restricts the application of EDLCs.

Presently, research on EDLCs is mainly focused on improving their energy efficiency and operating temperature range. Carbon materials are abundant and environmentally friendly, with high specific surface area, good electrical conductivity, high chemical stability, and a wide operating temperature range. These are promising electrode materials for supercapacitor applications. Various types of carbon materials are currently widely used as electrode materials in commercial EDLCs. The carbon electrode materials of supercapacitors have an efficient specific surface area, an optimal pore volume and pore size distribution, and good electrical conductivity and wettability.

#### 1.1.2. Pseudocapacitor

In 1951, Conway proposed the Faraday quasi-capacitor mechanism based on the two-dimensional or quasi-two-dimensional capacitor [13]. This model suggested an electrode charging potential mechanism via underpotential deposition with reversible adsorption–desorption redox reactions. For a Faraday quasi-capacitor, the charge storage process includes storage on the double layer and the redox reactions between electrolyte ions and the active materials. When the ions in the electrolyte (such as H^+^, OH^−^, K^+^, or Li^+^) diffuse from the solution to the electrode/solution interface under the action of the applied electric field, they will enter the bulk phase of the active oxide on the electrode surface through redox reactions at the interface, resulting in the storage of a large amount of charge in the electrode. During the discharge process, these ions entering the oxide will be returned to the electrolyte through the reverse of the above redox reaction, and the stored charge is released through the external circuit. This is the charge and discharge mechanism of the Faraday quasi-capacitor. Several pseudocapacitors with different charge storage mechanisms are shown on the left side of Figure 3. Pseudo-capacitors can store charge by electroporation, redox reactions, or intercalation, allowing them to have higher capacitance and energy density than EDLCs. It should be noted that pseudocapacitors do not have the same characteristics as typical batteries since they undergo surface redox reactions, in contrast with batteries, where redox reactions are conducted on the bulk of the electrode materials. When pseudocapacitors are repeatedly used for charging and discharging, they become depleted more rapidly than EDLCs.

Pseudocapacitance can be generated on the electrode surface and inside the entire electrode; thus, higher capacitance and a higher energy density can be obtained compared with EDLCs. With the same electrode area, the capacitance of pseudocapacitors can be 10–100 times the capacitance of electric double layers. In general, pseudocapacitors, categorized by metal oxides and conductive polymers, represent high resistivity, resulting in inefficient electron transport in the electrochemical process and low power density. Moreover, compared with the number of cycles (tens of thousands) of EDLCs, the cycle life of pseudocapacitors is poor. To reduce the interface contact resistance and increase the specific capacitance, the commonly used solution is nanometer electrode materials [14,15] or preparing electrode materials with special morphologies and structures to reduce the diffusion distance of electrolyte ions the conduction distance of electrons [16,17]. In addition, electrode materials can be viably grown on conductive substrates to prepare binder-free supercapacitors [18,19]. 

#### 1.1.3. Hybrid Capacitor

Hybrid capacitor technologies aim to achieve high performance by combining the advantages of EDLCs and pseudocapacitors. However, SCs have less energy density than batteries and fuel cells, which is a major challenge in the present scenario. Most recent developments in electrode material for SCs have focused on enhancing their energy density, which involves developing electrode materials with a very high specific capacitance (F/g) which enables cells to operate at higher voltages. Hybrid or asymmetric supercapacitors produce higher capacitance using an electrode made of EDLC and other pseudocapacitive material. The higher voltage applied to electrodes made from Faradaic material causes a wide operating potential window for high-performance applications. Hybrid electrode material, which utilizes both EDLC and Faradic charge storage advantages, also helps develop high-performance energy storage devices. 

### 1.2. Carbon Nanotubes

Carbon nanotubes (CNTs) are one of the carbon allotropes having a fullerene-based columnar nanostructure. Kroto and Smalley discovered fullerene in 1985, while CNTs were discovered by Iijima in 1991. CNTs have a one-dimensional (1D) structure, excellent conductivity, high porosity, and excellent mechanical and chemical stability in supercapacitors, so they are widely used as active electrode material [20,21]. CNTs are classified into single-walled carbon nanotubes (SWCNTs) or multi-walled carbon nanotubes (MWCNTs) according to their structural morphology. 

#### 1.2.1. Single-Wall Carbon Nanotubes (SWCNTS)

The specific surface area and pore size of the CNT electrode affect the electrochemical performance. SWCNTs have a larger specific surface area and better specific capacitance than MWCNTs [22,23,24]. Zhu et al. [25] presented a hierarchically well-ordered core–shell CNTs@NiCo-LDH nanotube structure (Figure 4). Unlike conventional devices, this unique structure exhibits excellent electrochemical properties and reasonable material density and structural design. The CNT within the core–shell structure effectively reduces the diffusion path of electrolyte ions and solves the problem of the slow ion transport rate of external NiCo-LDH. In addition, the internal structure of the ion exchange pores was developed to reduce the effect of volumetric shrinkage during multicycle charging and discharging. The CNT-based core–shell structure showed excellent specific capacitance from 1 A g^−1^ to 176.33 mAh g^−1^. In addition, the fabricated supercapacitor exhibits a high energy density of 37.38 Wh kg^−1^ at 800 W kg^−1^ and excellent cycling stability (90.2% retention after 5200 cycles).

Hsieh et al. [26] presented a CoMn_2_O_4_ (CMO) nanosheet electrode coated on carbon nanotubes grown on stainless steel mesh (SSM) as an electrode for supercapacitors using chemical vapor deposition. CNTs provide much three-dimensional space for growing CMO nanosheets and preventing conduction due to agglomeration. They created defects for the further electrodeposition of CMO nanosheets via chemical vapor deposition on SSM. Electrodeposited CMO shows an advantageous shape by controlling deposition time, and excellent specific capacitance performance can be obtained. Electrodeposited CMOs show the high oxidation potential of cobalt and the fast electron transport ability of manganese. CMO grown on CNT and SSM inhibits the use of binders or additives and prevents unwanted resistance. The binder-free CMO/CNT electrode exhibits an excellent specific capacitance of 732 F/g^−1^ at a scan rate of 2 mV/s. In addition, when an asymmetric supercapacitor (ASC) using such a hybrid network as a cathode and activated carbon (AC) as an anode was fabricated, energy density was 47.39 W/kg^−1^, power density was 400 Wh/kg^−1^, and power failed after 5000 cycles It showed good stability of 77% maintaining capacity. The equivalent series resistance (ESR) must be minimized for high-power supercapacitors. When pure CNTs are used as electrode materials for supercapacitors, a surface area corresponding to the holes stacked between CNTs is mainly used. In general, the internal surface area of CNTs is not usable, but the specific capacitance can be improved by modifying the internal surface. The most common method is to activate CNTs using acid or alkaline solutions. Previous studies have used 10 wt% nitric acids [27], a mixture of nitric acid and concentrated sulfuric acid (1:3) [28,29] or alkaline solutions such as KOH [30,31] or NaOH [32,33,34]. In this way, the specific surface area and pore volume were greatly improved. 

#### 1.2.2. Multi-Wall Carbon Nanotubes (MWCNTs)

A mechanism by which MWCNTs of different structures are activated with KOH and NaOH has also been reported [35]. The NaOH used was effective only in disordered structures, but KOH showed that it was effective in all structures. Starting with a redox reaction between KOH or NaOH, the chemical activation process produces metal K, Na, and metal carbonate (K_2_CO_3_ or Na_2_CO_3_). Frackoviak et al. [36] developed highly developed pure multi-walled carbon nanotubes (MWCNTs) using chemical KOH activation. Mesoporous nanotube materials are widely used electrochemically because of the microporosity of the surface. The researchers found that when the KOH/C ratio was 4:1, the activated material had a nanotube morphology with a greatly increased micropore volume while maintaining its mesoporous properties. These activated MWCNTs have been used as electrode materials for supercapacitors in acidic, alkaline, and aprotic media. As a result, the capacitance increased 7-fold from 15 F/g^−1^ for inactivated nanotubes to 90 F/g^−1^ through chemical activation. Gupta et al. [37] prepared a polyaniline(PANI)/single-wall carbon nanotube (SWCNT) composite through electrochemical polymerization and studied its capacitive and energy-power properties. Electrodes using the PANI/SWCNT composite showed much higher specific capacitance, specific energy, and specific power than pure PANI and SWCNT. SWCNTs decorated with 73 wt% PANI had the highest specific capacitance, specific power, and specific energy of 485 F g^−1^, 228 Wh kg^−1^ and 2250 W kg^−1^. It was confirmed that the desired specific capacitance could be obtained by controlling the microstructure of the PANI/SWCNT composite material.

The specific capacitance can be increased through chemical strengthening by adding redox-active functional groups to CNTs. However, excessive oxidation reduces conductivity and reduces cycle life. Xia et al. [38] fabricated PANI/CNT hybrid supercapacitors with core–shell nanostructures. The PANI/CNT core–shell hybrid showed high levels of 1005 F g^− 1^ in 1 M H_2_SO_4_ electrolyte at 2.0 A g^− 1^. An improved specific capacitance of 1547 F g^− 1^ was also achieved by adding Fe^3+^/. Using 0.02 M Fe^3+^/Fe^2+^ as redox, an improved specific capacitance of 1128 F g^− 1^ was obtained. The flexible supercapacitor fabricated with PVA/H_2_SO_4_/Fe^3+^/Fe^2+^ gel electrolyte provided an energy density of 22.9 Wh kg^−1^ at a power density of 700.1 W kg^−1^. Lee et al. [39] Composites were prepared by decorating the CNT surface with MoO_3_ and MnO_2_. In terms of specific surface area, MoO_3_@CNT improved to 68 m^2^ g^−1^ and MnO_2_@CNT to 343 m^2^ g^−1^, and the capacitance retention rate was maintained at 96.8% at 5 A g^−1^ even after 10,000 cycles. MoO_3_@CNT and MnO_2_@CNT composites are used as cathodes and cathodes, respectively, providing the possibility of developing safe and inexpensive energy storage devices with three-dimensional porosity and good conductivity.

Because of their appealing properties and the possibility that they will behave similarly to graphene, carbon nanotubes (CNTs) are becoming more common electrode materials in supercapacitor applications. Because of the small area of their atomic structures, carbon nanotubes (CNTs) are considered a very power-dense material. Other issues with CNTs causing contamination are also a barrier to their adoption; this can be addressed by adding steps to the manufacturing process to improve production quality. CNTs (or, in some cases, graphene) can be used to improve the conductivity and structure of activated carbon/other electrodes.

### 1.3. Graphene Derivatives

Graphene was first isolated in 2004 by Russian physicists Geim and Novoselov by attaching scotch tape to a pencil lead and repeatedly peeling off the graphite powder attached to the tape with glass tape. Graphene is a fascinating material with a wide range of potential applications. The hexagonal lattice of graphene is composed of a single layer of carbon atoms. Graphene has recently attracted tremendous attention due to its high specific surface area and excellent electrical conductivity. Notably, electrons can move ballistically in an sp^2^-bond layer of graphene at a speed of 15,000 m^2^V^−1^s^−1^ at room temperature without being scattered, opening a new field of study for electrochemistry applications [40]. There are several different graphene derivatives, each with unique properties. For example, graphene oxide (GO) is more chemically stable than graphene, making it ideal for certain applications. Similarly, reduced graphene oxide (rGO) is more electrically conductive, making it suitable for electronic devices. There are many potential applications for graphene and graphene derivatives. Due to these properties, graphene derivatives such as graphene oxide (GO) and reduced graphene oxide (rGO) are widely used in light-emitting diodes, touch screens, field effect transistors, solar cells, supercapacitors, batteries, and sensors [41,42]. In addition, graphene shows high energy storage ability by forming a complex with various electrode materials such as metal oxide, conducting polymer, carbon nanotube, and activated carbon. Graphene has the potential to be a suitable material for supercapacitors. This material has high conductivity and surface area, which make it ideal for storing and releasing electrical energy. Additionally, graphene is strong and stable and can withstand being charged and discharged many times without compromising its shape or integrity. Graphene has an extraordinarily high specific surface area of up to 2630 m^2^ g^−1^, has a very high theoretical capacitance (area) of up to 550 F g^−1^, and conducts electricity very well; these characteristics make it a good candidate for supercapacitors. Graphene electrodes that are less stacked have higher disorder, and plenty of available defect sites (edge planes) help enhance the capacitance of graphene supercapacitors. It is possible that graphene can be destacked, or at least stacked into a single sheet, but keeping them unstacked while assembling a device and applying the device is difficult. Gao et al. [43] synthesized an oriented three-dimensional graphene framework (3DGF) that resembles paper but with oriented surfaces, macropores, and interconnected parts created using hard template-directed ordered assembly. 3DGF shows a high specific surface area of 402.5 m^2^/g, which allows one to determine the number of pores, and is mechanically flexible for application as a supercapacitor electrode. The synthesis approach provides controlled pore size and the homogeneous laminar structure of polystyrene microspheres, which provides an efficient way to produce 3DGF with a high degree of control over the pore size and orientation (Figure 5). It has a high specific capacitance of 95 F/g at 0.5 A/g, allowing an enhanced rate capability. Yan et al. [44] developed graphene nanospheres (GNS) by combining template separation with microwave heating and graphitizing carbon layer to form hollow graphene nanospheres. Advancements in synthesized graphene helped to maximize performance further. GNS materials with a specific surface area of 2794 m^2^ g^−1^ can deliver capacitances higher than 529 F g^−1^ at 1 A g^−1^ and have capacitance retention of 62.5% in a continuous power supply. The graphene was developed with widened interlayer spacing, defects, and connected graphene bridges, which helped the electrodes deliver superior electrochemical performances [45]. Wen et al. [46] designed MXene-based composite films with different graphene content that were successfully printed onto a thin graphene film using inkjet printing. Graphene composite electrodes retain the excellent conductivity of the matrix itself, but they also exhibit large interlayer spacing, which reduces the transport path of electrolytes to get to the electrodes. The self-stacking effect of MXene is improved by incorporating graphene nanosheets into MXene structures. It is possible to achieve a very high capacitance of 183.5 F cm^−3^, and a supercapacitor assembled using those electrodes achieves an outstanding energy density of 0.53 MWhcm^−2^. To prevent the stacking effect that graphene sheets may have when stacked on top of each other, spacers such as metal oxides and conductive materials are inserted between the two layers of graphene to make them more accessible to electrolytes and other electroactive sites.

#### 1.3.1. Graphene Oxide (GO)

GO is a material that has been attracting much attention lately for its potential applications in many fields. GO is a single layer of carbon atoms that have been oxidized or had oxygen atoms added. This results in the addition of oxygen atoms to the carbon lattice. GO is made by exfoliating graphite in strong acids. The resulting material is a one-atom-thick sheet of carbon with oxygen atoms bonded to the carbon atoms on the sheet. One of the most notable things about graphene oxide is its flexibility. This makes it ideal for applications with important flexibility, such as electronic devices. Graphene oxide is more chemically stable than graphene, meaning it is less likely to break down over time. This stability makes it a promising material for long-term energy storage and other applications where durability is important. Graphene oxide is less conductive than graphene, but it is still a good conductor of electricity. Tian et al. [47] used a high-speed homogenous disperser to exfoliate a very large area of graphite and to add low-cost additives to make graphene/graphene oxide nanosheets. Prepared composites of graphene and graphene oxide nanosheets have better electrochemical properties (specific capacitance of 483 F g^−1^ at 1 A g^−1^). The electrochemical properties of the composite materials show that they perform well in electrochemical activity, which is ascribed to their 3D structure, high conductivity, high specific surface area, and homogeneity. Li et al. [48] designed and executed a study that evaluated the different degradation pathways resulting from hydrothermally reduced GO (htrGO)-based inhibitors. They systematically studied the different initial oxidation levels present in htrGO structures. The effects of the initial oxidation on the energy storage behavior of the material are demonstrated by analyzing its surface and interface properties. They prepared open framework structures that span different length scales that facilitate the conductance/transport of ions, increasing the density of oxygen-containing functional groups on the surface of these structures. The structure obtained by optimizing GO can deliver 330 F g^−1^ at 1 A g^−1^.

#### 1.3.2. Reduced Graphene Oxide (rGO)

GO is reduced graphene oxide (rGO) when the oxygen atoms are removed. This process can be achieved using various techniques, such as thermal reduction, chemical reduction, or electrochemical reduction. Thermal reduction is the most common method and involves heating GO in an inert atmosphere, such as nitrogen or argon. Sahu et al. [49] synthesized lacey-reduced graphene oxide nanoribbons (LRGONR) by chemically unzipping MWCNTs with the help of a strong oxidizing agent. It obtained high specific capacitance results from a combination of EDL and pseudocapacitance from oxygen-related surface groups in the edge planes. It was found that removing graphene layers with holes in them would enhance the electrolytic accessibility of the graphene but also cause it to become layered in order to create some deformation or twisting of the graphene. Because of the improved structural properties, LRGONR in H_2_SO_4_ has a very high energy density (15.06 W h kg^−1^) at a power density of 807 W kg^−1^. 

Recently, it has been possible to produce graphene or reduced graphene oxide (rGO) with the help of a few simple chemical reactions into a supercapacitor or other energy storage device materials. Restacking graphene/rGO layers by noncovalent interactions is a serious concern when developing electrolyte dispersion layer (EDL) capacitors based on synthesized graphene or rGO. Graphene and rGO that have not been chemically synthesized are hydrophobic and are not easily wettable. Graphene and rGO samples generated from synthesized chemically are hydrophilic and exhibit good wetting properties. Hydrophobic interactions between graphene and rGO are the main reason graphene/rGO layers are re-stacking [50]. Restacking neighboring layers can harm the capacitive performance of the capacitor because the electrolyte area available for electrolyte formation is decreased, and the ions in the solution are more strongly dispersed. When neighboring layers are restacked, it causes a decrease in the electrical conductivity of the capacitor due to the decreased effective area for the electrolyte and increased diffusion resistance for ions. Some efforts have been made to prevent the stacking of neighboring layers and to increase the capacitance of EDL-based capacitors. Functionalizing graphene/rGO with a hydrophilic functional moiety would prevent the formation of static charges and increase the wettability of the surface. Such functionalization can significantly enhance capacitance performance.

Graphene-based nanomaterials are not commercially available in large quantities needed for industrial applications. It should be noted that the performance depends on the purity and size of the graphite sheets, and the production methods that allow very high purity also have high costs and low yields. It is possible to mitigate this by reducing graphene oxide (as it is simpler to make and cheaper), which is easier to make in bulk (although the properties will still be affected). However, researchers are working to overcome those challenges. Despite the challenges associated with their use, scientists are striving hard to make them work efficiently. Graphene could be a key component of a new energy storage device. Graphene-based hybrid supercapacitors are very attractive to researchers because of their special properties. Researchers are working on improving the energy density for supercapacitor applications and reducing their costs. Future technology for graphene-based supercapacitors will be important in developing energy storage systems.

### 1.4. Activated Carbons

Among the various electrode materials for supercapacitors, activated carbon electrodes are well suited for the development of supercapacitors because they provide a large specific surface area, are stable in terms of physical and chemical properties, conduct electricity well, and are inexpensive. However, since the specific surface area of typical activated carbon material is about 1000–1500 m^2^ g^−1^, and the specific capacitance is low, so there is a limit to the application of ultra-high-capacity, electric double-layer polar materials. Therefore, the main focus of research to date is optimizing the porous structure, morphology control, and surface modification of activated carbon to obtain a higher specific capacitance. Activated carbon with a high specific surface area is generally prepared from carbonized organic precursors rich in carbon with an inert gas (heat treatment stage) and is selectively oxidized in carbon dioxide, water vapor, or KOH (activation stage) to obtain a high specific surface area and pore volume. After the activation step, the activated carbon has a porous network structure, and the average pore size of the activated carbon can increase with increasing activation time or temperature to improve the accessible surface area. The large adsorption capacity of activated carbon is attributed to the large specific surface area provided by the void structure. Multiple adsorptions can occur on the inner surface of macropores (>50 nm in size), and the adsorbed molecules enter the activated carbon through the macropores. The number of macropores mainly controls the adsorption rate. Mesopores (2–50 nm) have the same function as macropores and adsorb macromolecules that cannot enter micropores (<2 nm) [51]. The micropores of activated carbon are mainly responsible for the adsorption function and help to improve the electrochemical performance. However, if the pore diameter is too small, the internal resistance increases, slowing electrolyte movement within the pore.

Activated carbon can be derived from a wide range of raw materials. Common precursor materials for preparing activated carbon include wood materials (including all types of wood [52,53,54], bamboo [55,56,57], and starch [58,59,60]), waste from agricultural and food industries (such as coconut shell [61,62,63], walnut shell [64,65], cocoa shell [66,67], banana peel [68,69,70], peanut peel [71,72], rice husks [73,74], bagasse [75,76], and tea leaves [77,78]), coal-based materials (anthracite [79,80], bitumen [81,82], and lignite [83,84,85]), petroleum materials (petroleum coke [50,51,52,86,87,88], needle coke [53,54,55,89,90,91], pitch [56,57,58,92,93,94], and petroleum residue [59,60,61,95,96,97]), and plastic materials (phenolic resin [62,63,64,98,99,100] and poly(vinylidene chloride) [65,66,67,101,102,103]). Different precursor materials directly affect the structure of the prepared activated carbon. Moreover, the activation stage is particularly important in preparing activated carbon, which determines the pore diameter and specific surface area of the activated carbon and directly affects its electrochemical performance. The commonly used activation methods are physical or chemical. The physical activation method is also called gas activation. It uses water vapor, flue gas (the main component is CO_2_), oxygen-containing gas, or mixed gas such as air as the activator, and it is activated via contact with a carbonized material at a high temperature (above 600 °C) or activated by two activators. As a result, activated carbon products with a large specific surface area and well-developed pores are produced. Through the gasification reaction, the closed pores of the carbonized material are opened and enlarged, the pore wall is burned out, and certain structures are selectively activated to generate new pores. The formation of pores in activated carbon is closely related to the degree of carbon oxidation. Within a certain range, a deeper degree of gasification between the activated gas and carbonized material results in a larger specific surface area of the activated carbon producing more developed pores and better adsorption performance. The physical activation method involves a simple production process and is not subjected to problems such as equipment corrosion and environmental pollution. The prepared activated carbon can be directly used without washing.

In the chemical activation method, raw material and a concentrated solution of an activator are usually mixed and reacted. The mixed raw materials are dried, heated, and pyrolyzed in an activation furnace. The pyrolyzed product is extracted to remove the activator and obtain the activated carbon product. Commonly used activators are H_3_PO_4_ [104,105], H_2_SO_4_ [106,107], CaCl_2_ [108,109], NaOH [110,111], ZnCl_2_ [112,113], and KOH [114,115]. The chemical activation method can decompose and separate hydrogen and oxygen contained in hydrocarbons in the raw material in the form of water, thereby significantly reducing the carbonization temperature. Based on these two activation methods, microwave [116,117,118] and continuous carbonization [119,120,121] activation methods have been developed. These methods have higher thermal efficiency, allow the uniform heating of materials, resulting in less pollution, and require a low carbonization temperature.

The specific surface area is an important factor in determining the electrochemical properties of activated carbon. In the case of a constant surface area, the specific capacitance of activated carbon increases with an increase in its specific surface area [122,123,124]. Lu et al. [125] prepared mesoporous activated carbon for high-performance supercapacitors using corn straw. Corn straw-based activated carbon provides a high specific capacitance of 202, 188 F g^−1^ at a current density of 1 A g^−1^ in organic and ionic liquid electrolytes. This is much better than commercial activated carbon YP50 (51 Wh kg^−1^). Therefore, it can be seen that the carbon material with optimal mesopores through activation exhibits low-cost and high-performance characteristics as an EDLC material.

However, the theory that the specific capacitance of activated carbon increases with an increase in the specific surface area is not always accurate due to the ionic size effect (such as the ionic radius or Stokes radius) [126] and the surface-specific capacitance of the micropores and the outer surface is different [127]. Barbieri et al. [128] investigated the relationship between the specific capacitance and surface area of various carbon materials. It could be seen that the capacitance of the carbon material did not increase linearly as the specific surface area increased, and the capacitance tended to be stable when the specific surface area was 1200 m^2^ g^−1^ or more. Therefore, as the specific surface area increases, the pore walls cannot accommodate the same amount of charge at a given electrode potential, and the gravimetric capacitance saturates as the specific surface area increases.

One of the key factors determining the activated carbon material of EDLC is the pore size distribution. Dolas et al. [129] created activated carbons from physically activated pistachio shells after chemical activation. After impregnating the pistachio shells with ZnCl_2_, the HCl solution was impregnated to form a fine mesopore structure. The activated carbon with the largest specific surface area was produced from the pistachio shells impregnated with a 40% salt solution. The larger the pore size, the less energy is stored in the activated carbon, but the faster the energy can be transferred. Therefore, activated carbon with a large pore size is suitable for application in high-power supercapacitors. Furthermore, Talreja et al. [130]. reported phenol–formaldehyde–resin-based activated carbon with controlled carbon pore size. Metal ions (Fe^+^, Zn^+^) were used as templates, and the pore size was controlled in the range of 2–50 nm by changing the metal ions. Zn-carbon particles exhibited high specific capacitance of ion 152 Fg^−1^ due to high mesoporosity. The supercapacitor composed of Fe/Zn-carbon particles had a maximum energy density of 64 Wh kg^−1^ and a maximum power density of 709 kW kg^−1^. From this, it can be seen that activated carbon with controlled pore size distribution improves the fast diffusion of electrolytes and the performance of supercapacitors.

Generally, when microporousness is very high, it is difficult to pass electrolyte ions, resulting in excessive internal resistance. On the other hand, the very high macroporosity contributes to the small specific surface area and small specific capacity; thus, the pore size distribution directly affects the electrochemical performance of the material. Ordered mesoporous carbon (OMC) is a type of carbon material with a high specific surface area and porosity. The size of the pores can be adjusted within a certain range, and the mesopores have various shapes. Ordered mesoporous carbon (OMC) is a type of carbon material with a high specific surface area and a high porosity. The size of the pores can be adjusted within a certain range, and the mesopores have various shapes. These mesopores facilitate the rapid and large-scale transport of ions (Figure 6).

Du et al. [132] prepared ordered mesoporous carbons (OMCs) by converting phenol monomers to mesoporous carbons. OMC was prepared by magnetic deposition according to the solid-phase grinding method, and the obtained OMC replicated the morphology of the template and exhibited a high surface area, large pore volume, and a uniform mesoporous structure. The pore size of the mesopores is larger than the ionic size of the electrolyte, which is beneficial for the rapid transmission of ions and the maintenance of low internal resistance, thus improving the specific capacitance of the material. In addition, the specific capacitance of EDLCs is related to the mesoporous surface area but not the BET-specific surface area. Studies have shown that doping heteroatoms such as nitrogen [133,134] and boron [135,136] in the OMC structure can help enhance the properties of the carbon surface, further improving the electrochemical performance of the material. Ragavan et al. [137] prepared ordered mesoporous carbons (OMCs) doped with a series of heteroatoms such as nitrogen and phosphorus. Among the prepared samples, nitrogen and phosphorus co-doped OMC (NPOMC) exhibited a high specific capacitance of 355 F g^−1^ due to heteroatoms at a current density of 0.5 A g^−1^. Through this, the heteroatom-doped OMC structure improved the electrical conductivity and surface affinity of the carbon structure, and the OMC structure provided a channel for the fast mass transport of electrolyte ions.

A type of hierarchical porous carbon-containing micropores, mesopores, and macropores has been extensively studied. As shown in Figure 7, the carbon contains pores of different sizes, and many pore structures are interconnected and assembled in a hierarchical form. As mentioned previously, the presence of micropores provides a large surface area for enhanced charge storage capability. On the other hand, mesopores, macropores, and hierarchical structures can improve electrolyte penetration and promote ion diffusion. The capacitance of carbon-based supercapacitor electrodes is generally maintained between 100 and 200 F g^−1^, whereas the capacitance of graded porous carbon-based supercapacitor electrodes exceeds 300 F g^−1^.

Zhu et al. [138] applied a hierarchically porous carbon (HPC) material produced from heavy bio-oil to supercapacitors and found that the carbon material prepared with the NaOH/heavy bio-oil-derived carbon precursor ratio of 3:1 had the highest specific surface area at 2826 m^2^ g^−1^ and the largest pore volume at 1.78 cm^3^ g^−1^. The specific capacitance of a single HPC-3 electrode in a two-electrode symmetric supercapacitor reached 259 F g^−1^ at 0.5 A g^−1^ in 6 M KOH electrolyte. Here, it has a higher energy density and specific capacitance than many biomass-derived activated carbons. Yang et al. [139] synthesized hierarchically porous carbons (HPCs) with heteroatom-doped multiscale hybrid structures from carbonized low-cost petroleum asphalt. The synthesized HPC showed capacitance of 437 F g^−1^ in KOH electrolyte of 1 A g^−1^ and excellent rate performance of 336 F g^−1^ at 50 A g^−1^. In addition, HPC electrodes exhibit high electrochemical behavior in Na_2_SO_4_ and PVA/KOH gels and electrolytes, making them promising electrode materials for high-performance supercapacitors.

ACs exhibited a specific capacitance of 244.5 F g^−1^ at a current density of 0.2 A g^−1^ in 6 M KOH aqueous electrolyte. The capacity remained at 81.8% of the initial capacitance at 40 A g^−1^, and the specific capacity remained at 91.6% after 10,000 cycles. The device could provide an energy density of 8.5 Wh kg^−1^ at a power density of 100 Wh kg^−1^. SCs have widely adopted activated carbon due to its highly porous nature, ability to tailor pore size and distribution, and low cost of production, but restricted conductivity results from high porosity, which restricts ACs as high-performance supercapacitor applications.

### 1.5. Carbon Fibers 

Activated carbon fiber (ACF) has various advantages, such as high porosity, high volumetric capacity, and excellent packing density [140]. In general, carbon-containing organic fiber precursors are stabilized at a low temperature (200–400 °C) and then activated at a high temperature (700–1000 °C) to produce nanoscale pores on the surface and increase the specific surface area. An ACF has a fiber diameter of 5–20 μm, an average specific surface area of around 1000–3000 m^2^ g^−1^, and an average pore diameter of 1.0–4.0 nm, and the micropore volume accounts for more than 90% of the total pore volume. The channels of ACFs are relatively unobstructed, and the macropores, mesopores, and micropores are closely connected, which is beneficial for the transmission and adsorption of electrolyte ions. In comparison with granular activated carbon, an ACF has a larger adsorption capacity and faster adsorption kinetics. The specific capacity is 280 F g^−1^, and the specific power is greater than 500 W kg^−1^ [141,142]. Commonly used precursor fibers include pitch [143,144] rayon [145,146,147], phenolic resin [148,149,150], and polypropylene [151,152,153]. 

ACFs prepared from different raw materials have different electrochemical properties. Carbon fibers derived from environmentally friendly biomaterials have recently been the research focus. For example, Chen et al. [154] fabricated binder-free self-supporting supercapacitor electrodes using high-performance cellulose-based activated carbon fiber paper (Figure 8). The resulting activated carbon fiber paper (ACFP) is characterized by high specific surface area (808–1106 m^2^ g^−1^), high conductivity (1640–1786 S m^−1^), good tensile strength (4.6–6.4 MPa), and flexible processability. ACFP also exhibits a specific capacitance of up to 48.8 F cm^−3^ based on the entire electrode and excellent cycling stability. This demonstrates that paper-based electrode materials are widely applicable and cost-effective for energy storage applications.

The adsorption of an ACF to adsorbate molecules depends on its pore structure, surface characteristics, and the characteristics of adsorbate molecules. In order to achieve the desired performance, the ACF pore structure is usually adjusted, or its surface structure is functionalized. The electrochemical performance of an ACF can be improved by changing the activation process and degree of activation. Luo et al. [155] fabricated asymmetric supercapacitors using activated carbon fibers (ACF) composed of double hydroxides of Ni and Co layers and activated carbon extracted from fir bark as anodes and cathodes. NiCo-LDH aggregates around the ACF to form a microsphere-like core–shell structure. This structure provides a large surface area, hierarchical porosity and numerous active sites for efficient charge and mass transfer. The synthesized ACF showed the highest capacitance at 1 A g^−1^ to 1453.3 F g^−1^. In addition, the capacitance of 79.8% was maintained even after 10,000 cycles. Lin et al. [156] used oxidized polyacrylonitrile (PAN) fibers as precursors and investigated the effect of different activation methods (traditional chemical, traditional physical, and improved chemical) and different treatment temperatures (700, 800, 900, and 1000 °C) on the performance of a porous ACF electrode. Studies have shown that different activation methods affect the ACF porous structure, carbon layer stacking height, and capacitance performance. Improved chemical activation could increase the number of oxygen functional groups (such as C-O and -COOH) and the specific surface area of the capacitor. The ACF obtained a specific capacitance as high as 158 F g^−1^, which was higher than the specific capacitance obtained using the two conventional activation methods. In a certain temperature range, the specific capacity of the ACF electrode increased with an increase in the activation temperature; however, after reaching the peak value, the specific capacity decreased. In addition to the BET surface area and pore volume, surface oxygen functional groups also play an important role in capacitive performance.

ACFs have a higher density of unpaired electrons, have high reactivity, and can easily react with other elements to form chemical functional groups that dominate the surface chemical structure. Surface modification can improve the adsorption capacity of ACFs. Ni et al. [157] synthesized pitch-based oxygen–nitrogen co-doped porous carbon fibers with a high specific surface area using asphaltenes as the raw material. The prepared porous carbon fibers were used as electrode materials for supercapacitors in a three-electrode system. The as-prepared ACFs exhibited a high specific capacitance of 301 F g^−1^ and 482 F g^−1^ at 1 A g^−1^ in 6 M KOH and 1 M H_2_SO_4_, respectively. The improved electrochemical performance of ACFs may be attributed not only to the high specific surface area and abundant small micropores, which can provide more charge storage, but also to doping surface heteroatoms (nitrogen and oxygen), which can improve wetting behavior.

### 1.6. Carbon Aerogels 

Carbon aerogel is a new lightweight, porous, amorphous, bulk nanocarbon material [158]. Its continuous three-dimensional network structure can be controlled and tailored at the nanometer scale. It is an aerogel with a porosity of 80–98%, a pore size of less than 50 nm, a network colloidal particle diameter of 3–20 nm, a specific surface area of 600–1100 m^2^ g^−1^, and good electrical conductivity of 25–100 S cm^−1^ [159,160]. In the 1980s, the Lawrence Livermore National Laboratory in the United States obtained carbon aerogels using the polycondensation reaction of resorcinol and formaldehyde as the raw material for the first time. Li et al. [161] prepared a carbon aerogel with a pearl-like network structure through the polycondensation reaction of resorcinol (R) and formaldehyde (F) with sodium carbonate as the catalyst. The electrochemical performance of the carbon aerogel electrode in three types of electrolytes (6 M KOH, 1 M Na_2_SO_4_, and 2 M (NH_4_)_2_SO_4_) was evaluated. The experimental results showed that the carbon aerogel electrode was stable in the KOH electrolyte. The capacitance was approximately 110 F g^−1^. The maximum capacitance of supercapacitors using the carbon aerogel as the active electrode material was 28 F g^−1^.

As mentioned earlier, the ideal electrode material should have a hierarchical porous structure with large pores for ion buffer pools, mesopores for ion transport, and micropores for enhanced charge storage. Therefore, the porous carbon structure of a carbon aerogel should preferably have multiscale pores. Liu et al. [131] fabricated multifunctional supercapacitors using nanocellulose-based composite carbon aerogels (Figure 9). The manufactured nanocellulose-based carbon aerogels. Provides high compressibility and excellent fatigue resistance due to its high capacitance of 109.4 mF cm^−2^ at 0.4 mA cm^−2^ and its porous structure. The porosity of the honeycomb structure efficiently transferred stress to the entire microstructure and contributed to rapid ion transport. Because of these characteristics, 85% of the capacitance was maintained even after 10,000 cycles. The carbon aerogels produced through this process are supercapacitor electrodes with excellent mechanical flexibility and linear sensitivity, which can be widely applied to wearable devices.

In addition to the specific surface area, the pore size distribution, electrical conductivity, or power density could also affect the performance of supercapacitors. Yu et al. [163] prepared porous carbon aerogels with carboxymethyl cellulose as the raw material using the pyrolysis method, and KOH further activated the obtained carbon aerogels. The results showed that the KOH-activated carbon aerogels had a high specific surface area (428 m^2^ g^−1^), much higher than that of carbon aerogels without KOH activation (108 m^2^ g^−1^). Furthermore, the activated carbon aerogels had highly porous and interconnected three-dimensional nanostructures, which could provide channels for the migration of electrolyte ions and electrons. The activated carbon aerogels exhibited a high specific capacitance of 152.6 F g^−1^ at 0.5 A g^−1^ in 6 M KOH solution. Liu et al. [164] prepared a carbon aerogel using the resorcinol-formaldehyde method with sodium carbonate as the catalyst. The prepared carbon aerogel was subsequently activated with CO_2_, and the activation of CO_2_ significantly increased the specific surface area of the carbon aerogel. Under optimal activation conditions, a maximum specific surface area (3431 m^2^ g^−1^) was obtained. The maximum specific capacitance and energy density of activated carbon aerogel EDLCs were 152 F g^−1^ and 27.5 Wh kg^−1^, respectively, at a current density of 0.3 A g^−1^ with 1 M tetraethylammonium- tetrafluoroborate-ammonium (Et_4_NBF_4_) in acetonitrile as the electrolyte.

The surface modification of carbon aerogels also has a positive effect on electrochemical properties. Fang et al. [165] used surfactants to modify the surface of carbon aerogels. Vinyltrimethoxysilane (VTMS) functional groups were grafted onto the surface of activated carbon aerogels. The introduction of the surfactant VTMS increased the affinity of the carbon aerogels for propylene carbonate. The improvement in the surface wettability reduced the transport resistance of electrolyte ions in the micropores and increased the available surface area for EDL formation, thereby increasing the specific capacitance of the VTMS-modified carbon aerogels (specific capacitance enhanced by 41%; energy density increased by 274%).

Carbon-based high-performance supercapacitors are contributing significantly to the development of modern electronic devices. Carbon-based supercapacitors (carbon nanotubes, activated carbon, graphene, carbon fibers, carbon aerogels, etc.) have unique hierarchical structures, high specific surface areas, and excellent electrical properties that can be applied to autonomous driving sensors and wearable devices. Electrodes for carbon-based supercapacitors are summarized in Table 1 [52,53,56,93,99,133,163,166,167,168,169,170,171,172,173,174].

### 1.7. Other Carbon Materials 

In the last two decades, carbon and its hybrid materials have been extensively used in energy storage applications. Wood-derived carbon-based composites can be employed in a variety of applications. However, their electrochemistry is frequently subpar because of their tiny surface area and high impedance at the interface [175]. Carbonized wood has a distinct three-dimensional structure that can be utilized to support metal compounds and is also an ideal support for energy storage applications. Recently, it has been shown that combining heteroatom doping with the doping of carbon substrates can raise the electrochemistry performance of a hybrid electrode. Wang et al. [176] prepared hierarchically stacked wood-derived monoliths coated with Ag nanoparticles, and they manufactured NiCo_2_S_4_, which has a high areal/volumetry capacitance. This hierarchical structure of the WC@AgNiCo_2_S_4_ electrode shows excellent electrochemical characteristics, a capacitance of 6.09 Fcm^−2^, and long-term stability (84.5% stability) up to 10,000 cycles. This composite material can perform better than its predecessors due to a combination of factors, including the decreased adsorption energy of OH- and reduced d-band of Ag atoms, which help enhance electron transport and ion transfer.

Polyamide-derived carbon material is also a promising electrode material for supercapacitor applications; it also aids in the improvement of electrochemical performances by integrating with other nanomaterials. Polyimides (PI) with a dense, aromatic structure are a very interesting precursor for carbon compounds because of their facile and inexpensive synthesis and their good chemical and thermal stability [177]. Many N atom species in PI enable N-doped carbon skeletons to form when carbon is burned. Moreover, asymmetrical PI can be formed by utilizing pyridine molecules, which help form carbon materials with high yields and stability. Moreover, introducing some extra molecules, such as graphene and carbon nanotube, can modify the surface structure of PI. Li et al. [178] synthesized hierarchical porous structures of polyimide-derived carbon (CPC-Fe/Zn) electrode nanomaterials that are well-suited for hydrophilic use. This facile catalyst-based approach allows them to activate mixtures at high temperatures selectively. CPC-Fe/Zn can deliver a capacitance of up to 420 F/g at a current density of 1 A/g and has an excellent retention time of 92.5% even after 10,000 cycles. Bimetallics help to control the behavior of carbon electrodes by modifying their pore structure using this template-assisting technique Bimetalls help to control the behavior of carbon electrodes by modifying the pore structure of carbon electrodes. This provides a promising platform for many different applications. Further, chitin (obtained from various biomasses) has been widely used as an amino-functioning biomass precursor for synthesizing nitrogen-doping carbon using simple pyrolysis. Porous electrodes of chitin and chitosan-derived carbon are widely used in energy storage systems. They are highly accessible, highly porous, very light, and are naturally biodegradable, recyclable, and environmentally friendly. Zheng et al. [179] used chitin-derived nitrogen-doped porous carbons nanoparticles by allowing them to undergo a mechanically induced sol-gel transition in NaOH/Urea solvent and then carbonizing the porous nanoparticles with NaOH for activation and urea for N doping. The capacitance of a 3-electrode capacitance system is 245 F/g at a 0.5 A/g current density, and the capacitance of a 2-electrode capacitance system is 227 F/g with 98% retention after 1000 cycles. Recent research has demonstrated that flax is a low-cost, easy-to-prepare supercapacitor electrode material with good characteristics and high carbon content. Duan et al. [180] prepared zinc salt-treated flax fiber-derived carbon-based electrode materials for supercapacitors that were prepared by treating them with heat to impregnate them with zinc salt. They obtain specific capacitance is to be determined using 292 Fg^−1^ at 0.5 Ag^−1^ current density, and it remains at 102% retention with 10,000 continuous cycles.

Despite being exceedingly porous, porous carbon materials require numerous improvements in conductivity and electrochemistry. It is possible to increase the graphitization and conductivity of porous carbon materials by using catalyzing metallic species that act as anchor sites for other metallic species. The added structure of organic molecules such as carbon can give them higher reactivity and provide suitable sites for adsorbing metallic species. Doping heteroatoms with positively charged species changes the charge distribution in porously shaped carbon surfaces.

## 2. Carbon–Nanomaterial Hybrid Supercapacitors

### 2.1. Ceramic-Based Hybrid Supercapacitors

Most metal oxides used in pseudocapacitors are often hindered by the severe aggregation of nanoparticles, low electron–proton transport, and weak conductivity between nanoparticles, leading to a lower specific capacitance in practical applications. In order to effectively address these problems, the development of hybrid electrodes by combining metal oxides with carbon materials with a high specific surface area, such as activated carbon, graphene, CNTs, and carbon aerogels, has attracted widespread attention. Various metal oxide/carbon material composite electrodes for supercapacitors are summarized in Table 2 [181,182,183,184,185,186,187,188,189,190,191,192,193,194,195,196,197,198,199,200,201].

Because nanostructures cannot be synthesized on a large scale that allows for precise size control, their real-time application is limited [202]. In recent years, tremendous advances have been made in preparing ceramic nanostructures fabricated precisely by regulating the size, including nanorod and nanofiber structure. Carbon can be combined with ceramic nanomaterials to provide very high-energy density without sacrificing other performances (e.g., power density). Recently, Tiwari et al. [203] synthesized highly conductive MnS_2_/CNT and MnO_2_/CNT core–shell heterostructures by combining two different techniques: chemical and magnetic sputtering. With the contrast in work function between MoS_2_ and MnO_2_, they explore the maximum potential window that can be gained by implementing the asymmetric design. Self-standing prepared electrodes, which are 3-electrode configurations, provide very high capacitances of 0.41 and 0.6 Fcm^−2^, respectively, for MoS_2_/CNT and MnO_2_/CNT. The MoS_2_/CNT electrode has electrostatic polarization between −0.6 and 0.2 V, whereas the MnO_2_/CNT electrode has non-Faradaic charge storage between 0 and 1 V. By combining chemical and physical deposition techniques, this scalable synthesis technique allows for a massive array of edge-exposed catalytic sites available for electrode–electrolyte interaction. Hu et al. [204] synthesized NiCoO_2_ nanosheets using CNTs to bond them together using the hydrothermal method and fabricated integrated NiCoO_2_@CNTs@NF electrodes. These self-supporting electrodes have some advantages, such as having a structured 3D and very strong network, good conducting properties, and many sites where ions can be trapped and react quickly using a fast Faradaic redox reaction. NiCoO_2_@CNTs@NF-integrated electrodes have excellent capacitance and stability due to the outstanding synergistic effect between NiCoO_2_ nanosheets as Faradaic pseudocapacitance materials and CNTs as EDLC materials. The fabricated device has excellent performance because it possesses a high capacitance (151 F g^−1^ at 5 mA cm^−2^) and outstanding capability to operate at a high rate. Guo et al. [205] developed a binary network of carbon aerogel/Ni cubic carbon electrodes with thickness- and shape-independent properties. It was achieved using a very efficient and economical route consisting of the facile polymerization of technical lignin and formaldehyde in hypersaline conditions. The porosity and degree of graphitization and carbon residues in LCAN are controlled by the ratio of ZnCl_2_ to lignin. By precisely adjusting the ratios and combining the advantages of the LCA/Ni binary network, they thusly synthesize a cubic electrode with optimal electrochemical performance. Zhou et al. [206] used radish as a cheap catalyst and developed a facile method for synthesizing 3D carbon aerogels incorporating MnOx nanoparticles. The combination of MnOx nanoparticles and carbon nanotubes into a supercapacitor electrode material can prove beneficial, as carbon nanotubes allow for the easy transport of ions, thus enhancing the capacity of the supercapacitor. The electrochemical performance of the carbon aerogel/MnOx composite can be enhanced by the synergistic effect between carbon nanotubes and MnOx nanoparticles. Prepared carbon aerogel-based electrodes (CAE) exhibit the highest gravimetric capacitance (GP) of 557 F g^−1^ at a current density of 1 A g^−1^ in a 3-electrode system. Carbon and pseudocapacitive materials generally have high specific capacitance and high energy density and are expected to be incorporated with carbon aerogels to produce composite materials with better electrochemical performance. The combination of carbon aerogel energy density, specific capacitance and electrical conductivity could produce composite materials with higher electrochemical performance. Kumar et al. [207] used a simple and fast microwave approach to synthesize the composite rGO@CoO within a short duration of only 90 s for developing an advanced supercapacitor electrode. The hybrid composite combines the advantages of its capacitance properties with enhanced electrochemical performance, including higher specific capacitance and excellent long-term cycling stability. Transition metal oxides with various morphologies demonstrated promising electrochemical performance along with the graphene materials through synergistic contribution from each component as well as easy electrolyte ion insertion without distortion of the nanostructure.

### 2.2. Conductive Polymer-Based Supercapacitors

Conductive polymers will undergo noticeable swelling and shrinkage during charging and discharging, which could deteriorate the mechanical properties and cycle stability of the conductive polymer electrode. Nevertheless, a combination of conductive polymers and carbon materials can improve the chain structure, conductive properties, and mechanical stability and reduce costs, thereby increasing the cycle stability and specific capacitance of the composite. 

Liu et al. [208] designed high-performance supercapacitor electrodes using polypyrrole/polydopamine-modified carbon foam. As shown in Figure 10, independent and thick electrodes were fabricated by electrochemical deposition of polypyrrole and polydopamine on carbon foam with a hierarchical pore structure (CFF). In this structure, the interconnected and multilayered macropores of CFF effectively promote ion transport inside the electrode, and the addition of polydopamine reduces the stacking density of polypyrrole and increases the active area. As a result, when the deposition mass of the polypyrrole/polydopamine coating was 8.5 mg, the electrode capacitance achieved 1920 mF cm^−2^ at 1 mA cm^−2^ and an energy density of 0.12 mWh cm^−2^ at a power density of 0.22 mW cm^−2^. In addition, the capacitance was maintained at 100% even after 10,000 galvanostatic charge–discharge cycles.

Cherusseri et al. [209] prepared brush-like supercapacitor electrodes derived from oxidized carbon nanotube (OCNT) and polypyrrole (PPy). OCNTF, as a conductive substrate and current collector, provides a large specific surface area for the polymerization of PPy. The optimized OCNT/PPy represents the gravimetric capacitance of 305 F g^−1^, the gravimetric energy density of 42 Wh kg^−1^, and the maximum volumetric capacitance of 14,950 mF cm^−3^ at a current density of 2.5 mA cm^−2^. The enhanced pseudocapacitive property and specific capacitance are attributed to oxygen-containing functional groups and redox-active PPy of the OCNT/PPy electrode.

Due to the strong π-π stacking effect of CNTs, it easily undergoes severe aggregation, leading to the poor performance of supercapacitors, which can be improved by introducing conductive polymers, metal oxides, metal sulfides, etc. Zhu et al. [210] reported using Ag-doped PEDOT in PSS/CNT composites for thin-film all-solid-state supercapacitors. The introduction of Ag-doped PEDOT: PSS improved the conductivity of CNT electrodes, providing a high specific capacitance of 85.3 F g^−1^ (corresponding to 64 mF cm^−2^). In addition, the compact aligned CNT electrodes were tightly bound to the polyelectrolyte chain, and the assembled all-solid-state supercapacitors had a high tensile capacity of up to 480%.

Conducting polymers have a low charge–discharge rate due to their slower diffusion, which limits their power capability. Many of these polymers are weak and do not cycle very long because of their stresses during each cycle. The cycle lives of these devices can be greatly extended by adding carbon, and the efficiency of their electrochemistry can be greatly improved. These hybrid material electrochemical performances can significantly enhance more than pure polymers.

### 2.3. MOFs-Based Hybrid Supercapacitors

Metal–organic frameworks (MOFs) have attracted much attention due to a diverse range of advantages, including large surface areas, favorable pore-size distribution, superior stability, excellent tunable strength, and flexibility for various applications [211]. Some purely MOF materials have been used for supercapacitors, but their low conductivity and poor stability usually limit these. The poor stability of MOF materials is due to poor bonding; unlike covalent bonds in many materials, MOFs are formed by reversibly forming coordination bonds, forming metal ions that act as co-reactive linkers that form coordination bonds with other ions. There have been three ways to improve the conductivity and performance of MOF materials: (i) exploiting new conductive MOFs, (ii) developing different multifunction MOF-related composite materials, or (iii) obtaining the MOF-sourced materials. When MOF materials are incorporated into composites, the combined effect of the different materials can amplify their advantages and show new physicochemical properties that enable them to overcome their limitations. The MOF-derived metal oxide composites can produce some synergistic effects that individuals cannot provide and have gained more and more attention for SCs. Many recent studies have focused on combining conducting polymers such as polyaniline, PEDOT, PPy, and MOFs to prepare materials that can provide very good electrochemistry performance for supercapacitor applications. Further, increasing the number of MOFs in supercapacitors by preparing carbon-based materials such as AC, graphene, or CNTs can promote their charge transfer properties. Applications of carbon–MOF hybrid material to electrodes also faced some obstacles because the strong interaction between adjacent nanosheets leads to the large-scale self-restoration and aggregation of these sheets, severely limiting the access to electrolyte ions and damaging the electrochemistry of the devices. However, the potential of incorporating MOFs and metal oxides as SCs electrodes has not been fully explored. Compositing metal oxide layers on MOF-based electrodes will allow investigators to produce electrodes with better conductivity. This is due to the synergy effect between metal oxide layers and MOFs. It will be far better to design fabrication strategies for metal oxide/MOF nanocomposites by incorporating metal oxides into the MOF systems.

## 3. Challenges and Prospects

The advancement of supercapacitors has been a topic of great interest in recent years. Supercapacitors are a promising technology for a variety of applications. They have high power density, high capacitance, high energy, long cycle life, and no memory effect. However, some challenges are still associated with their development, including finding a suitable electrolyte and achieving high energy density. Future developments in supercapacitors are based on developing and applying novel electrode materials, combined with other materials to form a supercapacitor [212]. Packaging and sealing systems for energy storage devices must be improved greatly to enable them to be used as intended. Supercapacitors that resist extreme conditions, such as high and low temperatures, strong acids, and bases, are very important. This research will focus on developing new electrode materials and electrolytes and separating materials and binders. There is much work to be done to commercialize novel electrode materials. There has been a rapid increase in the number of flexible, miniaturized, lightweight devices with excellent electrochemistry performance devices that prompts the rapid development of flexible energy storage devices in modern electronics [213]. SCs that combine advantages such as being flexible and achieving high efficiency are suitable for such devices.

SCs are widely used in numerous energy, automotive, industrial, and consumer electronics applications. They can improve current batteries in many ways, such as providing power to devices in remote or difficult-to-access locations or providing power to smart devices that do not always need power. Supercapacitors have one drawback, however, which is their cost. However, new supercapacitor materials are being developed with the development of nanotechnology, nanomaterials, and nanofabrication. The current research on supercapacitors is focused on improving electrochemical performance, material characteristics, and fabrication. Advancements in these areas can lead to the greater performance of supercapacitors, thus improving their efficiency and cost-effectiveness. Further research is required to identify new materials; achieve a controllable structure; reduce internal resistance; and increase the capacitance, energy, and power density, and cycle life. Supercapacitors could significantly impact the transportation and energy sectors if these challenges can be addressed. One challenge is improving the energy density of supercapacitors. This is important because it directly affects the amount of power stored in a given device. Researchers are working on new materials and designs that could significantly improve energy density. Another challenge is reducing the self-discharge rate of supercapacitors; this is the rate at which the stored energy is lost over time. If this rate could be reduced, it would greatly improve the lifespan of the fabricated supercapacitor device. Figure 11 summarizes in a tabularized list possible supercapacitor enhancement strategies and challenges. As we see, the optimal architecture for supercapacitor design depends on factors such as cell structure, electrode material, fabrication, geometry, electrode materials, and electrolyte composition. The electrode material also determines the ultimate performance of a supercapacitor. For example, a carbon electrode with a graphene layer will exhibit higher specific energy than a carbon electrode without graphene but with lower volumetric capacitance. Finally, the electrolyte composition can determine how the electrode material interacts with it. The tabularized list has three sections. The first section describes the possibilities for the enhancement of supercapacitor performance. The second section contains the required strategies for achieving it. The third section contains supercapacitor enhancement challenges. The third section contains a detailed analysis of the challenges associated with enhancing supercapacitors. The goal is to identify areas where improvement is needed to make these devices more efficient and effective. Additionally, this section provides potential solutions to address these challenges.

Supercapacitor technology research’s main trend is increasing supercapacitors’ energy and power density. Here we discuss the latest advances in electrode materials and electrolytes for supercapacitors and how these materials can be optimized to improve performance.

### Challenges and Optimization Strategies for Improving Energy Density and Power Density of Supercapacitors

1.The energy density of a supercapacitor is the amount of energy that can be stored in the device per unit volume. This is an important parameter when choosing a supercapacitor for a particular application. Higher energy density ($E=1/2\;CV^{2}$) means that more energy can be stored in a given volume, making the supercapacitor more compact.

For supercapacitors to have a higher energy density, it is necessary to perform the following three things:i.Improve the specific capacitance (*C*) of the material or develop new electrode materials with high specific capacitance.ii.Enlarge the voltage window (*V*) of the supercapacitor.iii.Design and optimize hybrid batteries/supercapacitors or symmetric supercapacitors. Asymmetric supercapacitors can fully exploit the different potential windows of the two electrodes to maximize the voltages that can be applied to the capacitor. These kinds of devices can effectively increase the energy density of devices.

2.Power density describes the rate performance of energy storage devices. As can be seen from Figure 12, compared with other energy storage devices, supercapacitors show higher power density [214]. From the formula of power density ($P=V^{2}/4R$, where *P* is power density, *V* is the potential window, and *R* is the equivalent series resistance), it can be seen that equivalent series resistance (ESR) and voltage window (*V*) have a direct effect on power density. Furthermore, the voltage window not only has an effect on the energy density but also on the power density.

Supercapacitors are a promising technology for energy storage, but the electrode materials and electrolytes limit their performance. In addition, the energy density of supercapacitors is still much lower than that of batteries. This is due to the limited specific capacitance of electrode materials and the low solubility of electrolytes in the organic solvent. One way to optimize supercapacitor performance is carefully selecting the materials used for the electrodes and electrolytes. High-performance devices can be achieved by optimizing electrode materials and electrolytes. (Figure 13). In this section, we will discuss how to optimize electrode materials and electrolytes in order to improve the energy density of supercapacitors.

Several other factors can affect the capacitive performance of supercapacitors:(1)Nanomaterials are used to reduce the size of particles and increase the specific surface area of supercapacitors. Nanomaterials can have different morphologies or contain different effective pore sizes. Functional groups improve the surface wettability of electrode materials with the electrolyte, improving the cycle life.(2)The aqueous electrolyte has the advantages of being very concentrated, having a small ion radius, and having low resistance. Water can decompose at just 1.23 V, greatly limiting the device’s energy and power density. When using an electrolyte containing an organic compound, the voltage range extends to about 2.5–2.7 V. By increasing the purity of the electrolyte, it is possible to reduce the influence of impurities on the device’s working voltage range and cycle life.(3)Electrochemical resistance (ERS) describes the resistance of an electrode and its interface with a current collector. It also determines how well an electrode conducts current. ESR is determined by the area and porosity of the electrode, the conductivity of the electrolyte and separator, and the operating temperature. Electrodes are usually made by applying an electrically active material, conductive agent, and binder to a slurry that is applied to a current collector.(4)The operating temperature has a significant effect on the supercapacitor. Its capacitance will decrease under high temperatures, especially in the continuous high-temperature operation state. In addition, the working temperature greatly influences the viscosity of the electrolyte and the ionic conductivity.

Supercapacitors are effective energy storage devices in many industries and fields. The massive market will provide limitless prospects for the development of supercapacitors. However, there is still tremendous room for advancement in these beneficial energy storage technologies. Driving electric vehicles and wearing smart clothing are becoming increasingly trendy. As a result, it is important to broaden the application range of supercapacitors, lower costs, and boost energy density. However, there are still issues with some electrode materials that affect the performance and cost of supercapacitors. Figure 14 presents an overview of the state of efforts to obtain high-performance and highly applicable supercapacitor devices.

## 4. Outlook and Conclusions

The supercapacitor has shown great potential as a new high-efficiency energy storage device in many fields, but there are still some problems in the application process. Supercapacitors with high energy density, high voltage resistance, and high/low temperature resistance will be a development direction long into the future. The search for next-generation electrode materials and electrolytes for supercapacitors is an intensely active area of research. Optimizing electrode materials and electrolytes is critical to developing high-performance supercapacitors with improved energy density, power density, and cycle life. 

In general, research can start from the following four points. First, develop electrode materials with high performance, pore size distribution, specific surface area, electrical conductivity, surface functional groups, crystal structure, wettability, and conductivity; these have a great influence on the performance of the supercapacitor. Second, develop functional electrolytes suitable for high-voltage and/or high-/low-temperature conditions. In this process, the influence of the nature of the electrolyte salt, the viscosity of the solvent, the ionic conductivity on the voltage window, and performance of the supercapacitor should be considered. Third, build a hybrid battery–supercapacitor to solve the problem of the low energy density of supercapacitors. Fourth, in addition to the above improvements for electrodes and electrolytes, optimize the matching of electrodes–electrodes and electrodes–electrolytes during device assembly, which includes the matching of positive and negative work functions, the optimization of the reaction kinetics of the electrode/electrolyte interface, and the widening of the working voltage of the device by controlling the surface charge.

With the rapid development of electronic technology, flexible, wearable electronic micro-devices have received great attention in recent years. However, there are technical and economic limitations. For this reason, the active materials used in typical supercapacitors are mostly supplied in powder form due to their structural characteristics and are manufactured in a cylindrical cell shape. It is difficult to solve the issues of flexibility and electrode implementation. For a flexible energy storage device, it is necessary to study the application of powder-type active material to fiber-type energy storage cells that can be fabricated by methods such as knotting, twisting, and weaving. Compared with batteries, the energy density of flexible supercapacitors is too low, and the battery life is short, which is difficult for meeting long-term use needs in actual life. The dynamic mechanical deformation process places higher requirements on the flexibility of the electrode and the binding force of the surface multilayer materials. It is necessary to continuously develop electrodes with excellent electrochemical performance and good mechanical flexibility. In addition, the search for low-cost, green, and environmentally friendly electrode materials, as well as the development of simple and efficient flexible electrode preparation processes that do not require high equipment, are also issues that need to be urgently solved. The flexible supercapacitor matched with portable electronic products will be a potential development direction of the next generation of flexible storage devices.

Supercapacitors help achieve better energy conservation and emission reduction in automobiles, rail transit, and renewable energy power generation and have broad development prospects. In the future, reducing costs and enhancing performance will be the only way to accelerate the application of supercapacitors in a wider field.

## Figures and Tables

**Figure 1 nanomaterials-12-03708-f001:**
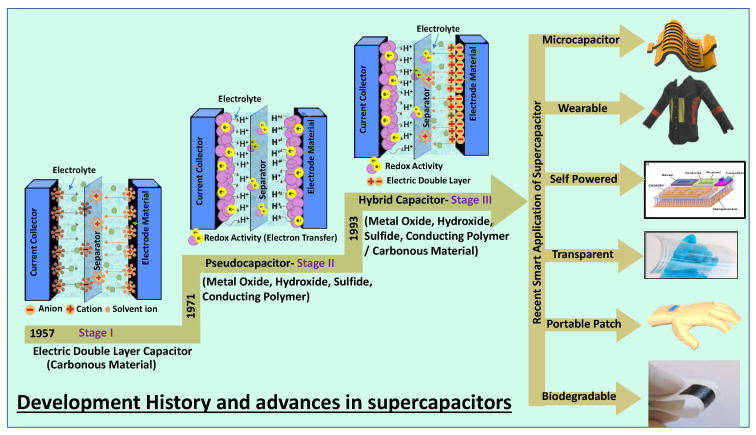
Classification of supercapacitors based on various electrode materials and their advanced applications.

**Figure 2 nanomaterials-12-03708-f002:**
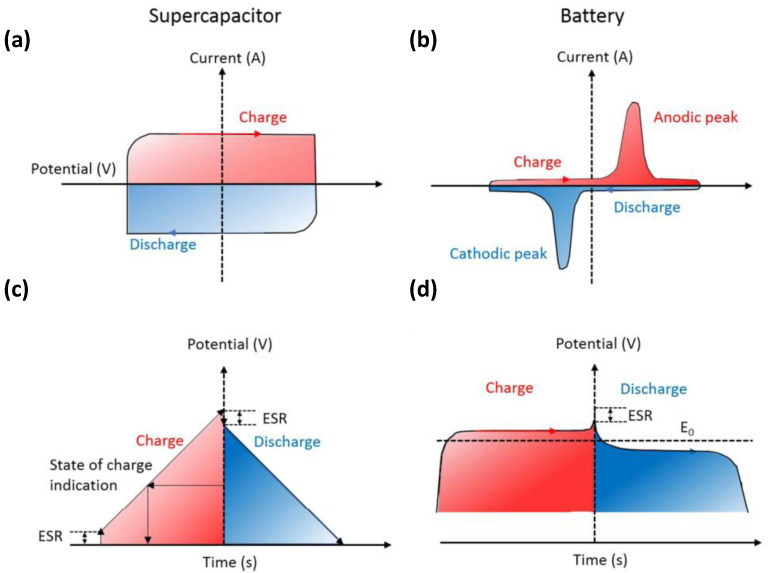
(**a**,**b**) cyclic voltammetry (CV) curves and (**c**,**d**) galvanostatic charge-discharge (GCD) curves of supercapacitors and batteries. Reproduced with permission [10] Copyright © 2022, Chemical Reviews.

**Figure 3 nanomaterials-12-03708-f003:**
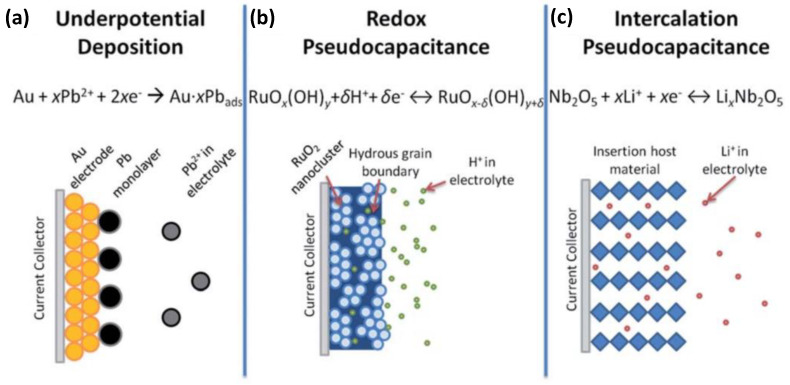
Schematics of the charge storage mechanisms for (**a**) EDLCs and (**b**–**d**) different types of pseudocapacitive electrodes: (**b**) underpotential deposition pseudocapacitor, (**c**) redox pseudocapacitor, and (**d**) ion intercalation pseudocapacitor. Reproduced with permission [11]. Copyright © 2022, American Chemical Society.

**Figure 4 nanomaterials-12-03708-f004:**
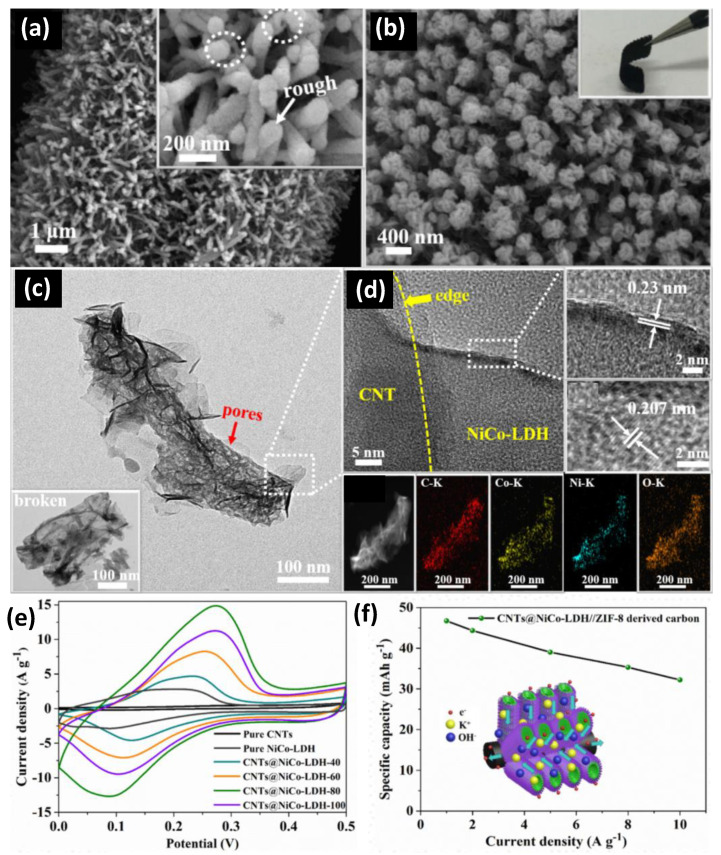
(**a**) ZIF-8 derived CNT arrays. (**b**) CNTs@NiCo-LDH core–shell nanotube arrays. (**c**) TEM image of CNTs@NiCo-LDH core-shell nanotube arrays. (**d**) HRTEM images of the as-synthesized CNTs@NiCo-LDH core-shell nanotube arrays and Elements mapping. (**e**) Typical CV curves of the CNTs@NiCo-LDH core-shell nanotube arrays at 5 mV s^−1^. (**f**) Specific capacity of the as-prepared CNTs@NiCo-LDH//ZIF-8 derived carbon ASC at different current densities. Reproduced with permission [25] Copyright © 2022, Chemical Engineering Journal.

**Figure 5 nanomaterials-12-03708-f005:**
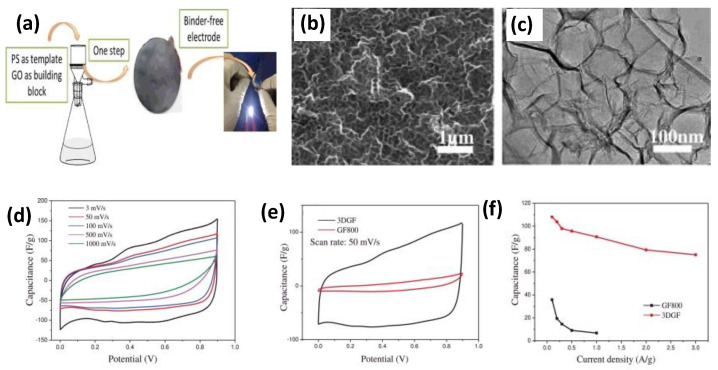
(**a**) Schematic illustration of binder-free electrode preparation. (**b**) Low-magnification SEM micrograph of 3DGF surface. (**c**) TEM image of 3DGF (high magnification). (**d**) CV curves of 3DGF at various scan rates. (**e**) Comparative CV curves of 3DGF and GF800 at 50 mV/s scan rate. (**f**) Calculated specific capacitance of 3DGF and GF800 at various current densities. Reproduced with permission [44] Copyright © 2022 Elsevier B.V. All rights reserved.

**Figure 6 nanomaterials-12-03708-f006:**
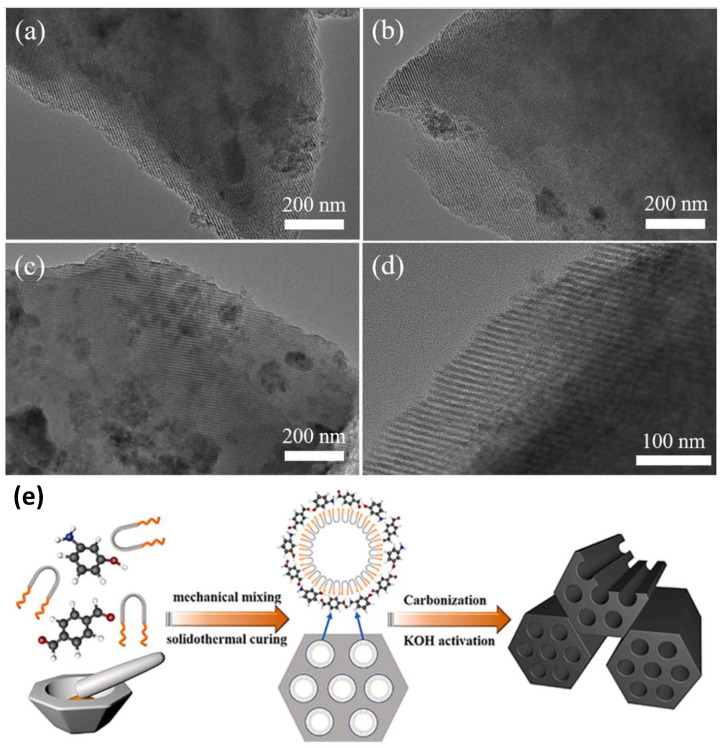
(**a**–**d**) Solidothermal synthesis of nitrogen-decorated ordered mesoporous carbons with large surface areas was analyzed by TEM characterization, (**e**) Synthesis scheme of K-N-OMC-T. Reproduced with permission [131]. Copyright © 2022, Chemical Engineering Journal.

**Figure 7 nanomaterials-12-03708-f007:**
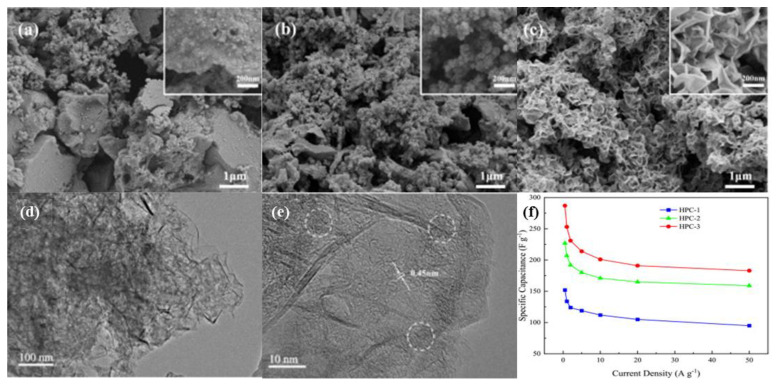
SEM images of (**a**) pore structure of hierarchical porous carbon (HPC-1) (**b**) foam-like pore structure of hierarchical porous carbon (HPC-2), (**c**) layered structure of hierarchical porous carbon (HPC-3), (**d**,**e**) TEM images of HPC-3 (**f**) The plot of the Gravimetric specific capacitances of HPCs at different current densities. Reproduced with permission [138] Copyright © 2022, Journal of Energy Storage.

**Figure 8 nanomaterials-12-03708-f008:**
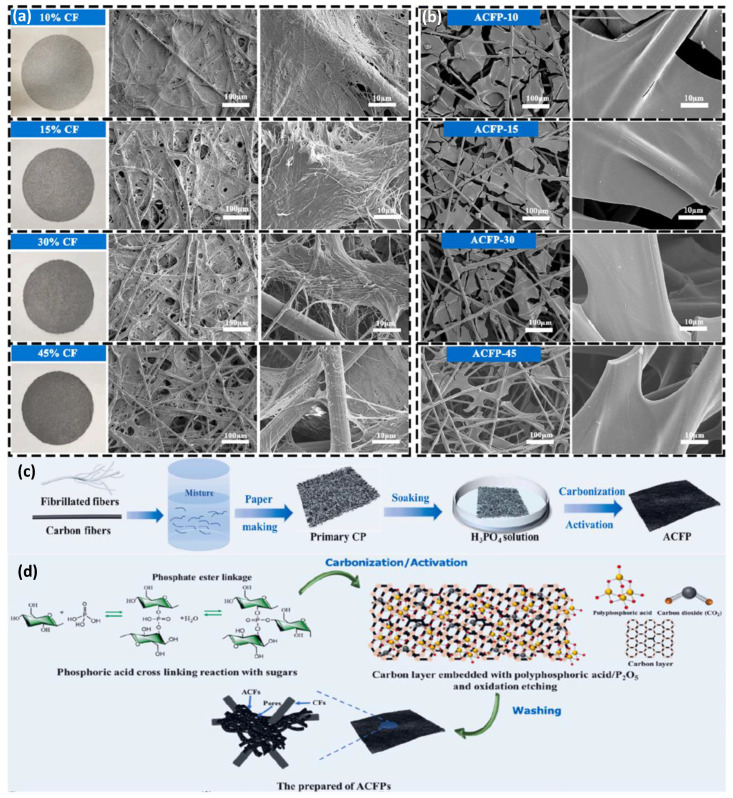
(**a**) Photographs and SEM images of primary CPs with different magnification (10% CF, 15% CF, 30% CF, 45% CF). (**b**) SEM images of cellulose-based ACFPs with different magnification (ACFP-10, ACFP-15%, ACFP-30, ACFP-45). (**c**) Schematic illustration of the fabrication of cellulose based ACFPs. (**d**) The process and mechanism of ACFP structure and pore formation. Reproduced with permission [154] Copyright © 2022, Chemical Engineering Journal.

**Figure 9 nanomaterials-12-03708-f009:**
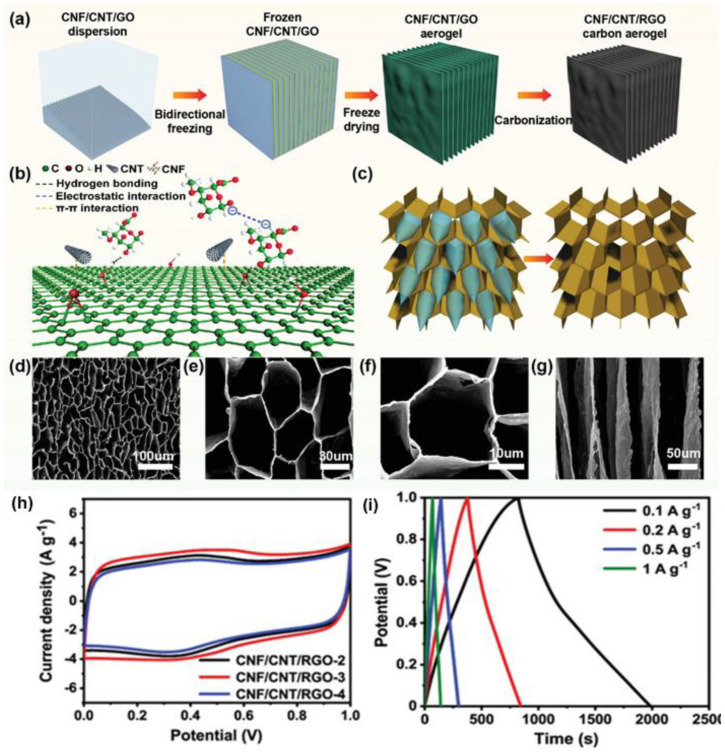
(**a**) Illustration for CNF/CNT/RGO carbon aerogels fabrication. (**b**) The schematic diagram of the interaction between CNF, CNT, and RGO. (**c**) Schematic diagram of ice crystal growth mechanism. (**d**–**f**) Top-view and (**g**) side-view SEM images of CNF/CNT/RGO-3 carbon aerogel. (**h**) CV curves of CNF/CNT/RGO-2, CNF/CNT/RGO-3, and CNF/CNT/RGO-4 at 2 mVs^−1^. (**i**) GCD curves of CNF/CNT/RGO-3 carbon aerogel. Reproduced with permission [162] Copyright © 2022, Advanced Functional Materials.

**Figure 10 nanomaterials-12-03708-f010:**
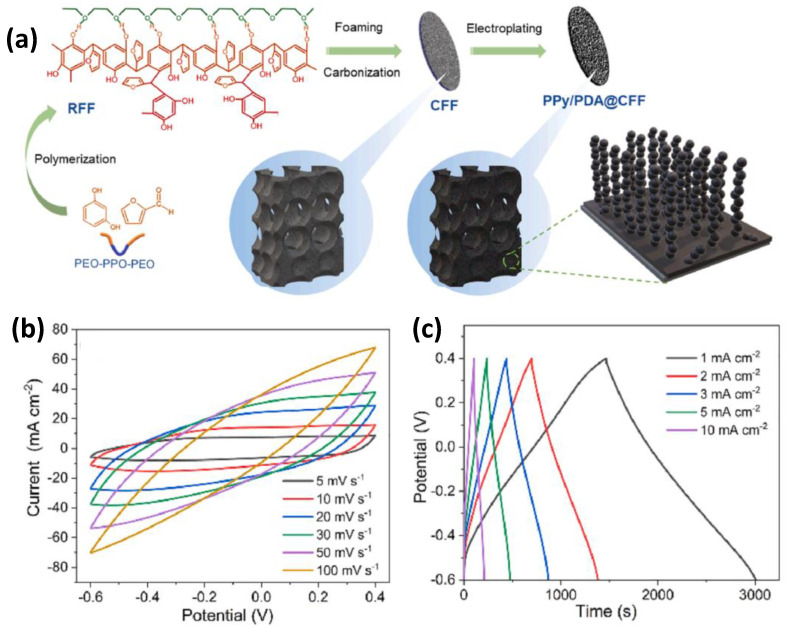
(**a**) Diagram of the preparation process of PPy/PDA@CFFs. (**b**) CV curves of PPy/PDA@CFF-20 at different scan rates. (**c**) GCD curves of PPy/PDA@CFF-20 at different current densities. Reproduced with permission [208] Copyright © 2022, Chemical Engineering Journal.

**Figure 11 nanomaterials-12-03708-f011:**
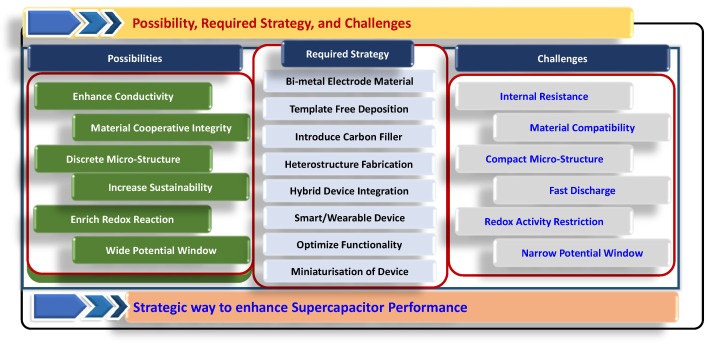
Tabularized list of possible supercapacitor enhancement strategies and challenges.

**Figure 12 nanomaterials-12-03708-f012:**
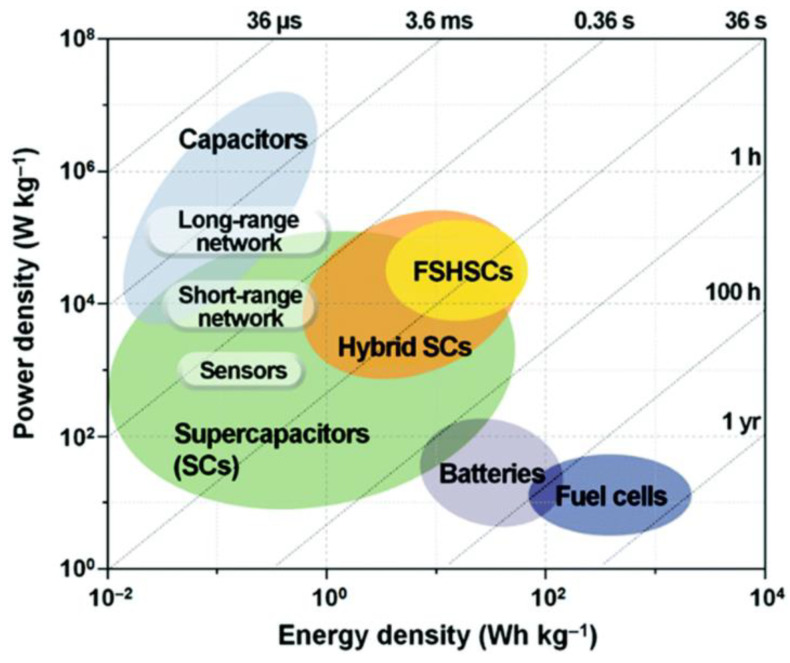
A Ragone plot of the specific energy and power densities of various energy storage devices. Reproduced with permission [7] Copyright © 2022, Energy & Environmental Science.

**Figure 13 nanomaterials-12-03708-f013:**
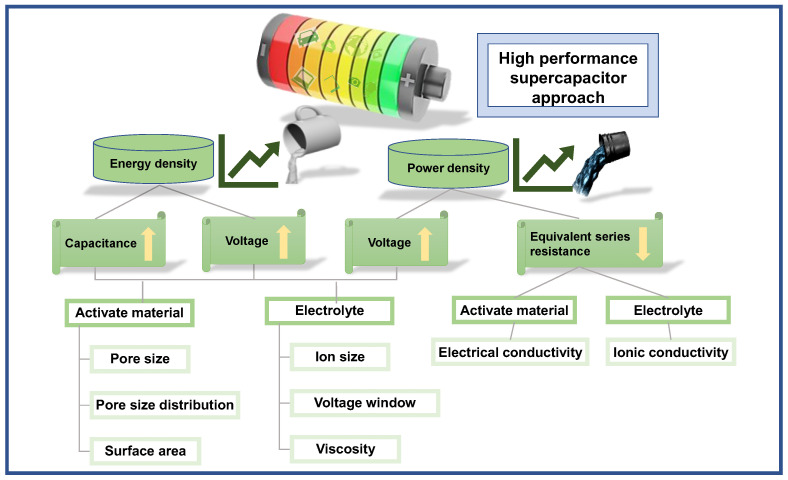
Schematic of a multifaceted approach to obtain high-performance supercapacitors.

**Figure 14 nanomaterials-12-03708-f014:**
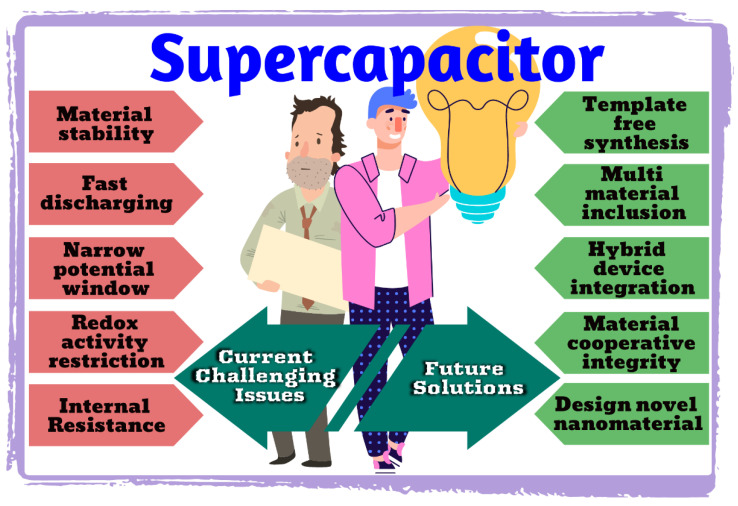
Schematic of several current challenges and future solutions to obtaining highly applicable supercapacitor devices.

**Table 1 nanomaterials-12-03708-t001:** Summary of the performance of carbon-based electrodes for supercapacitors. Electrodes for carbon-based supercapacitors are summarized in Table 1.

Material	Potential Window/V	Electrolyte	Specific Capacitance/F g^−1^ (Scan Rate or Current Density)	Retention/% (Cycles)	Ref.
Hierarchical porous carbon/KOH-activated wood sawdust	−1–0	6 M KOH	303 (1 A g^−1^)	99 (5000)	[52]
Activated carbon fiber/liquefied wood	−1–0.9	1 M H_2_SO_4_	280 (0.5 A g^−1^)	99.3 (2000)	[53]
Heteroatoms (O, N)-doped porous carbon	−1–0	1 M H_2_SO_4_	223 (1 A g^−1^)	93.6 (4000)	[56]
Pomegranate-like porous carbon	−1–0	6 M KOH	341.3 (0.1 A g^−1^)	96.1 (5000)	[93]
Rice husk (RH)-derived AC	0–1	6 M KOH	330 (0.5 A g^−1^)	92 (2000)	[99]
Walnut shell based activated carbon (ACWS)	0–1	6 M KOH	330 (0.1 A g^−1^)	95 (10,000)	[133]
Coal-tar pitch (CTP)/sawdust (SD) co-carbonization	−1–0	6 M KOH	251 (0.1 A g^−1^)	93 (7000)	[163]
Nitrogen-doped ordered mesoporous carbon	−1–0.7	1 M H_2_SO_4_	264 (0.5 A g^−1^)	86 (10,000)	[166]
Boron-doped ordered mesoporous carbon (B-OMC)	0–0.9	1 M H_2_SO_4_	290 (20 A g^−1^)	–	[167]
Nitrogen-doped microporous carbon spheres (NMCSs)	−0.2–0.8	6 M KOH	416 (0.2 A g^−1^)	96.9 (10,000)	[168]
Nitrogen-doped hierarchical carbon (NHPC)	0–0.9	6 M KOH	257 (0.5 A g^−1^)	90.3 (10,000)	[169]
O-N-S co-doped hierarchical porous carbons	−1–0	6 M KOH	244.5 (0.2 A g^−1^)	91.6 (10,000)	[170]
Sisal-derived activated carbon fibers (SC)	−1–0	6 M KOH	415 (0.5 A g^−1^)	93 (10,000)	[171]
Biobased nano porous active carbon fibers	0–1	6 M KOH	225 (0.5 A g^−1^)	85.3 (10,000)	[172]
Hierarchical porous carbon aerogels	−1–0	6 M KOH	142.1 (0.5 A g^−1^)	93.9 (5000)	[173]
KOH-activated carbon aerogels	−1–0	6 M KOH	152.6 (0.5 A g^−1^)	–	[174]

**Table 2 nanomaterials-12-03708-t002:** Summary of various metal oxide/carbon composite electrodes for supercapacitors.

Material	Potential Window/V	Electrolyte	Specific Capacitance/F g^−1^ (Scan Rate or Current Density)	Retention/% (Cycles)	Ref.
ZrO_2_ carbon nanofibers	0–1	6 M KOH	140 (1 A g^−1^)	82.6 (10,000)	[175]
RuNi_2_O_4_/rGO composites	0–1	0.5 M Na_2_SO_4_	792 (1 A g^−1^)	93 (10,000)	[176]
NiO/activated carbon composites	0–0.4	2 M KOH	568.7 (0.5 A g^−1^)	90.6 (5000)	[177]
Ni_0.25_Co_0.25_oxide/carbon nanofibers	−1–0	6 M KOH	431.2 (1 A g^−1^)	94 (2000)	[178]
MnO/Fe_2_O_3_/carbon nanofibers	0–1	6 M KOH	437 (1 A g^−1^)	94 (10,000)	[179]
ZnO/MnO/carbon nanofibers	0–1.6	6 M KOH	1080 (1 A g^−1^)	96 (800)	[180]
Au-Mn_3_O_4_/GO nanocomposites	−0.2–1	0.5 M H_2_SO_4_	475 (1 A g^−1^)	94 (10,000)	[181]
Bi_2_O_3_/MWCNT composites	−1.2–0.2	6 M KOH	437 (1 A g^−1^)	88.7 (3000)	[182]
NiO/MnO_2_/MWCNT composites	0–0.55	2 M KOH	1320(1 A g^−1^)	93.5 (3000)	[183]
Carbon nanosheets/MnO_2_/NiCo_2_O_4_ composites	0–1	1 M KOH	1254 (1 A g^−1^)	81.9 (5000)	[184]
ZrO_2_/C nanocomposites	0–1	1 M H_2_SO_4_	214 (1.5 A g^−1^)	97 (2000)	[185]
NiO/porous amorphous carbon nanostructure	0–1.6	6 M KOH	508 (1 A g^−1^)	78 (3000)	[186]
Defective mesoporous carbon/MnO_2_ nanocomposites	−0.8–0.8	1 M Na_2_SO_4_	292 (0.5 A g^−1^)	79 (2000)	[187]
Activated carbon/MWCNT/ZnFe_2_O_4_ composites	−0.1–0.6	3 M KOH	609 (1 A g^−1^)	91 (10,000)	[188]
NiO/C@CNF composites	−0.1–0.5	3 M KOH	742.2 (1 A g^−1^)	88 (5000)	[189]
N-doped carbon quantum dots/Co_3_O_4_ nanocomposites	−0.4–0.6	6 M KOH	1867 (1 A g^−1^)	96 (500)	[190]
Mn_3_O_4_/Fe_3_O_4_@Carbon composites	−0.4–1.2	1 M NaCl	178 (1 A g^−1^)	95 (1000)	[191]
rGO/CNTs/MnO_2_ composites///	0–1.8	1 M Na_2_SO_4_	332.5 (0.5 A g^−1^)	89.2 (10,000)	[192]
(HPC)/polyaniline (PANI) nanowire	0–1.8	1 M H_2_SO_4_	1080 (1 A g^−1^)	91.6 (5000)	[193]
Cyclodextrin polymer-functionalized polyaniline (CDP)/porous carbon composites(PC)	−0.2–0.8	6 M KOH	437 (0.1 A g^−1^)	81 (5000)	[194]
MnO_2_/Graphene Oxide/Polyaniline composites	0–1	1 M Na_2_SO_4_	512 (0.25 A g^−1^)	97 (5100)	[195]

## Data Availability

Not applicable.

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
