# Peer review of "Recent Advanced Supercapacitor: A Review of Storage Mechanisms, Electrode Materials, Modification, and Perspectives"

_nanomaterials, 2022, doi:10.3390/nano12203708_

Round 1
Reviewer 1 Report
The authors presented here a decent review article on the recent progress in the field of supercpacitors along with challenges and prospective. Few comments/suggestions are:
(1) The focus of the REVIEW ARTICLE is not clearly presented. It's very vaguely described instead of developing on a specific topic in mind. The authors have tried to cover a wide range of subject on supercapacitors, with storage mechanisms, type of electrode materials along with material modification resulting in inconsistency among subsections and not a very comprehensive study. Recommended, the author must make it more focussed.
(2) Although the review is very consistently written, there are too much generic descriptions in the introduction and some other sections making it quite redundant and not providing with any new information that are not documented or reported before. For example, "...CNTs are one-dimensional quantum materials, forming specific 169 spatial structures and sp3 hybrid bonds by sp2 hybrid bonds between two carbon atoms. 170 CNTs have a high surface area, low mass density, superior electrical conductivity and 171 mechanical strength, good corrosion resistance, and chemical stability...." Such generic and routine information must not be a part of the REVIEW ARTICLE. Recommended, the author must explain their intended approach in a more scientific way.
(3) Contemporary literatures are well cited but not comprehensively reported. The purpose of the REVIEW ARTICLE is to present specific and scientific comparison and findings. Recommended, the authors must identify trends in types of electrode materials and provide a comparative view rather than discussing individual literature reports and increasing length of the review article.
(4) Please consider using better reproduced images, especially in Figure 2 & 3. Their quality is not upto the mark.
(5) The REVIEW ARTICLE must be more precise. The purpose of some figures are not clear. Recommended, the authors must club figures together instead of just copying whole images of other authors. Each figure must be presented to serve a purpose.
(6) The authors have mentioned comprehensive description of three types of SCs development in their abstract, but the information provided for these three types are only limited to their generic storage mechanism description in the introduction section. It does not give enough understanding in terms of their development timeline and recent status. Recommended, the authors must make separate subsections or change the focus of the review.
(7) Some of the references had incorrect format or they are not consistent with each other. Please review.
Reviewer 2 Report
The paper can be accepted after answering the questions
The manuscript deals about the recent advances in supercapacitors. The article is well written but needs major revision before acceptance.
1. Supercapacitor technology has been explored for many years, and there are several review publications in the field. The underlying mechanism and scientific discussions appear to be less, despite the fact that the authors provide a variety of examples under each type of carbon, the scientific discussions, that are crucial for the review article to stand out, is less.
2. Maybe the author should make sure to explain “what has been done” and “what has to be done” for each section.
3. Authors may consider including recent research works on MOFs and other materials (Journal of applied physics,2007,102,111101) Could elaborate on the metal organic framework-derived carbon, which is the recent hot topic. This reference, which deals with supercapacitors should be quoted in the revised manuscript.
4. Recently developed composite materials and MOFs are less discussed, which must be elaborated with the underlying mechanism discussion.
5. Overall the grammatical errors and formatting must be corrected
Reviewer 3 Report
This review presents three types of superconducting capacitors, electrochemical double-layer capacitors (EDLCs), pseudo capacitors and hybrid supercapacitors. Their respective development, energy storage mechanism, and the latest research progress in material preparation and modification are reviewed. In addition, the possible feasible solutions to the problems encountered in the development of supercapacitors are proposed, and the future development direction of supercapacitors is prospected. However, there are still some issues to be addressed. This reviewer suggests a minor revision before its publication in this journal.
1. For a good review, a good overview diagram is necessary for the guidance in introduction part. In addition, in conclusion and outlook, one more important figure is required to highlight the present issues and future solutions.
2. Some more precursors for the preparation of electrode materials should be listed, introduced and discussed in the section 2, for examples:
1) Chitin: Journal of Bioresources and Bioproducts, Volume 6, Issue 2, May 2021, Pages 142-151
2) Wood: Polymers 14 (13), 2521, 2022; Wood Science and Technology 56, 1191–1203, 2022
3) Polyimide: Diamond and Related Materials 128, 109283, 2022; Journal of Colloid and Interface Science 609, 179-187, 2022
4) Flax fibers: Diamond and Related Materials 129, 109339, 2022
5) Camellia Pollen: ChemNanoMat 7, 202000531, 2021
6) Grape: Chinese Chemical Letters 31 (7), 1986-1990, 2020.
7) Etc.
3. The published data is discussed good but more critical evaluation is required in a review paper. Please update it.
4. To have a better readability and understanding, more figures should be added into the main text. In addition, some figures should be modified to have a better readability, especially the texts.
5. As a review article, comprehensive summary on highly related review articles is really necessary: Nano Today 37, 101075, 2021; Journal of Materials Science 56, 173–200, 2021; Journal of Materials Chemistry A 8, 23059-23095, 2020
6. When introducing the Pristine carbonaceous supercapacitor, the key points in "2.1. Carbon nanotubes" can be further divided according to the topography. Authors should carefully consider the description of these sections.
7. When generally introducing hybrid capacitors, recent review articles should be referred: Wood‐Derived, Conductivity and Hierarchical Pore Integrated Thick Electrode Enabling High Areal/Volumetric Energy Density for Hybrid Capacitors.
8. There are still some typos and grammar issues. Authors should check the whole manuscript again.
Round 2
Reviewer 2 Report
The manuscript can be accepted for publication as such
Reviewer 3 Report
Authors have made well revision according to the previous comments. An acceptance is suggested.